# Protein-inspired antibiotics active against vancomycin- and daptomycin-resistant bacteria

Mark A.T. Blaskovich [1], Karl A. Hansford [1], Yujing Gong[1], Mark S. Butler [1], Craig Muldoon[1], Johnny X. Huang[1], Soumya Ramu[1], Alberto B. Silva[1,5], Mu Cheng[1], Angela M. Kavanagh[1], Zyta Ziora [1], Rajaratnam Premraj[1], Fredrik Lindahl[1], Tanya A. Bradford[1], June C. Lee [1], Tomislav Karoli [1,6], Ruby Pelingon[1], David J. Edwards[1], Maite Amado[1], Alysha G. Elliott [1], Wanida Phetsang[1], Noor Huda Daud[1], Johan E. Deecke[1], Hanna E. Sidjabat[2], Sefetogi Ramaologa[1], Johannes Zuegg [1], Jason R. Betley[3,7], Andrew P.G. Beevers[3,8], Richard A.G. Smith[3,9], Jason A. Roberts[2,4], David L. Paterson[2] & Matthew A. Cooper[1]

The public health threat posed by a looming 'post-antibiotic' era necessitates new approaches to antibiotic discovery. Drug development has typically avoided exploitation of membrane-binding properties, in contrast to nature's control of biological pathways via modulation of membrane-associated proteins and membrane lipid composition. Here, we describe the rejuvenation of the glycopeptide antibiotic vancomycin via selective targeting of bacterial membranes. Peptide libraries based on positively charged electrostatic effector sequences are ligated to *N*-terminal lipophilic membrane-insertive elements and then conjugated to vancomycin. These modified lipoglycopeptides, the 'vancapticins', possess enhanced membrane affinity and activity against methicillin-resistant *Staphylococcus aureus* (MRSA) and other Gram-positive bacteria, and retain activity against glycopeptide-resistant strains. Optimised antibiotics show in vivo efficacy in multiple models of bacterial infection. This membrane-targeting strategy has potential to 'revitalise' antibiotics that have lost effectiveness against recalcitrant bacteria, or enhance the activity of other intravenous-administered drugs that target membrane-associated receptors.

[1] Institute for Molecular Bioscience, The University of Queensland, St. Lucia, QLD 4072, Australia. [2] UQ Centre for Clinical Research, The University of Queensland, Royal Brisbane and Women's Hospital Campus, Brisbane, QLD 4029, Australia. [3] AdProTech Ltd, Chesterford Research Park, Saffron Walden, Essex CB10 1XL, UK. [4] School of Pharmacy, The University of Queensland, Brisbane, QLD 4102, Australia. [5] Present address: AC Immune SA, EPFL Innovation Park, CH-1015 Lausanne, Switzerland. [6] Present address: Novasep (Dynamit Nobel Explosivstoff und Systemtechnik), Kalkstrasse 218, 51377 Leverkusen, Germany. [7] Present address: Illumina Cambridge Ltd, Capital Park, Fulbourn, Cambridge, CB21 5XE, UK. [8] Present address: Sterling Pharma Solutions, Sterling Place, Dudley, Cramlington, Northumberland NE23 7QG, UK. [9] Present address: School of Immunology and Microbial Science Kings College London, Guy's Hospital, London, SE1 9RT, UK. Correspondence and requests for materials should be addressed to M.A.T.B. (email: m.blaskovich@uq.edu.au) or to M.A.C. (email: m.cooper@uq.edu.au)

Antibiotic resistant bacteria pose a grave threat to human health and there now is an urgent need to develop new antibiotics[1–5]. Methicillin-resistant *Staphylococcus aureus* (MRSA), a major cause of community and hospital-acquired infections, leads to significant morbidity and mortality[1, 6, 7]. Vancomycin is often used to treat MRSA infections, but clinical isolates that are resistant are increasingly common[6]: VISA – vancomycin-intermediate *S. aureus*: MIC (minimum inhibitory concentration) 4–8 µg mL$^{-1}$; hVISA – heteroresistant VISA: predominantly susceptible but with a resistant subpopulation; and, more rarely[7], VRSA – vancomycin-resistant *S. aureus*: MIC ≥ 16 µg mL$^{-1}$. Even moderate reductions in vancomycin susceptibility (MIC shift from ≤1 to 1.5–2.0 µg mL$^{-1}$) can lead to significantly worse clinical outcomes[6, 8, 9]. In the last 15 years antimicrobial agents such as daptomycin and linezolid have emerged as approved therapeutic compounds for vancomycin-resistant bacteria[10], but resistance to both of these antibiotics was identified in vancomycin-resistant enterococci (VRE) and MRSA shortly after their approval[11, 12].

Drug development traditionally focuses on optimising target affinity, selectivity and protein binding in the context of drug pharmacodynamics[13], but rarely assesses membrane-binding properties[14]. In contrast, nature exploits subtle changes in the composition of biological membranes to control signal transduction, vesicular transport and receptor recycling with membrane-associated proteins such as Ras[15], Src[16], Rab[17], HIV-1 GAG[18] and MARCKS[19]. In these proteins, *N*-terminal and/or *C*-terminal lipophilic groups (myristoyl, palmitoyl, geranyl, farnesyl) and clusters of basic amino acids act as an electrostatic 'switch' to control translocation between the cytosol and membrane surface via selective phosphorylation of serine and tyrosine residues[16, 20, 21]. We postulated that appending such motifs to drugs acting on membrane-associated targets could enhance membrane binding and concomitantly increase drug concentration at the target site[20]. The potential also exists to discriminate between different types of membranes to increase drug selectivity. Given the alarming rise of multi-drug resistant bacteria and paucity of new antibiotics in the pipeline, we validated this approach using the glycopeptide antibiotic vancomycin.

In this study, we designed and synthesised a series of vancomycin derivatives, designated vancapticins, by appending membrane targeting motifs onto the *C*-terminus of vancomycin. The vancapticins display Gram-positive activity, including activity against vancomycin-resistant strains. The vancapticins were highly efficacious in multiple infections models in mice, with a low propensity for innate and induced resistance, and a pharmacokinetic (PK) profile consistent with once daily dosing in humans. Taken together, these results suggest vancapticins could be candidates for further development as a promising new therapy for the treatment of antibiotic-resistant Gram-positive bacterial infections.

## Results

**Chemical design of a membrane targeting motif.** Vancomycin blocks cell wall synthesis by binding to the *C*-terminal tripeptide (L-Lys-D-Ala-D-Ala) of the peptidoglycan precursor Lipid II, preventing cross-linking by bacterial transglycosylases and transpeptidases. A 'back-to-back' dimerisation enhances antimicrobial activity via a surface-templated chelate effect with membrane-bound Lipid II[22, 23]. In developing a membrane-targeting vancomycin analogue, it was important not to occlude the peptide binding site, nor disrupt interactions at the dimer interface. Previous derivatisations of vancomycin have utilised the vancosamine sugar amine or heptapeptide free carboxyl group;[24] other modification sites are possible, but more synthetically challenging. In order to attach basic electrostatic effector peptide segments containing lipophilic membrane-insertive elements (MIE), using chemistry compatible with unprotected functionality, we amidated vancomycin with pyridyldithioethylamine, then selectively ligated unprotected basic peptides containing a *C*-terminal Cys residue under mild conditions via disulphide exchange (Fig. 1, Y = S–S; Supplementary Figs. 1–3). The initial membrane targeting glycopeptides employed a lipophilically acylated 16-mer electrostatic effector peptide sequence (EEPS) (GSSKSPSKKKKKKKPGD)[25, 26]. These conjugates were fourfold to tenfold more active than vancomycin and daptomycin against MRSA, VISA and multi-drug resistant (MDR) *Streptococcus pneumoniae*, and more than 100-fold more active than vancomycin against VanA *Enterococcus faecium* (VRE) (Table 1). The corresponding non-ligated lipophilic peptides alone were not active, while equimolar mixtures of the same peptides with vancomycin displayed no synergy (Table 2). For the disulphide-linked series, maximal activity across most strains was observed with a dodecanoyl (MIE = nC11CO) insertive element, with potency dropping for both shorter and longer alkyl chain elements (compare **6**, **7**, **9** in Table 1). A biphenyl insertive element **10** provided comparable activity to the best alkyl chains. Progressive shortening of the initial 16-mer electrostatic effector

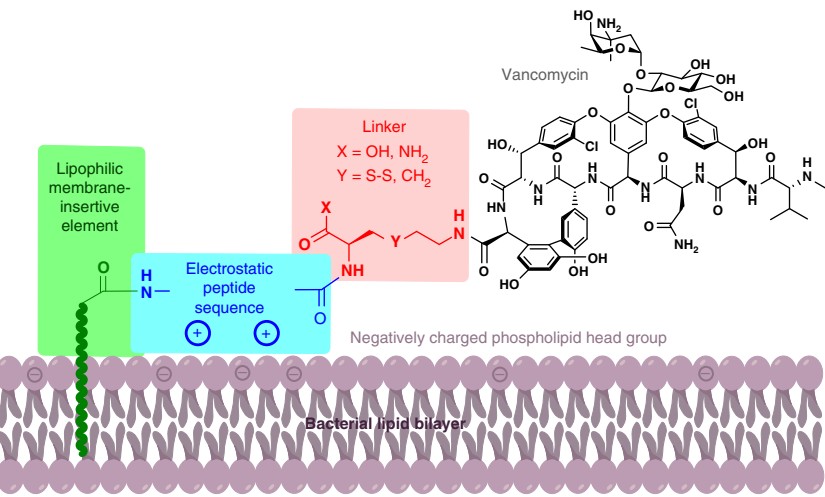

**Fig. 1** Structure of lysine-linked and cysteine-linked vancomycin-peptide conjugates. MIE membrane-insertive element, EEPS electrostatic effector peptide sequence

**Table 1 Structure and MIC of key compounds**

| | Structure | | | MIC (µg mL$^{-1}$) | | | | | |
|---|---|---|---|---|---|---|---|---|---|
| Id | MIE[a] | EEPS[b] | Linker | MRSA[c] | +50% HS[d] | VISA[e] | VRSA[f] | MDR S.P.[g] | VRE[h] |
| 1 | | Vancomycin | | 1 | 1 | 4 | >64 | 1 | >64 |
| 2 | | Telavancin | | 0.03 | 0.5 | 0.25 | 1 | 0.06 | 2 |
| 3 | | Dalbavancin | | 0.06 | 8 | 1 | 2 | 0.09 | >8 |
| 4 | | Daptomycin | | 2 | 3 | 12 | 3 | 2 | 16 |
| 5 | | Oritavancin | | 0.06 | 0.5 | 1 | 0.125 | 0.06 | 0.125 |
| *Cys-based disulphide linker* | | | | | | | | | |
| 6 | nC9CO[i] | GSSKSPS(K)$_6$PGD | C-NH$_2$ | 2.5 | >8 | >8 | nd | 3 | 1.25 |
| 7 | nC11CO[i] | GSSKSPS(K)$_6$PGD | C-NH$_2$ | 0.09 | >8 | 1 | nd | 0.06 | 0.02 |
| 8 | nC13CO[i] | GSSKSPS(K)$_6$PGD | C-NH$_2$ | 0.5 | >8 | 4 | nd | 0.5 | 0.125 |
| 9 | nC15CO[i] | GSSKSPS(K)$_6$PGD | C-NH$_2$ | 0.06 | >8 | 2 | nd | 0.125 | 8 |
| 10 | 4-Ph-Bz[j] | GSSKSPS(K)$_6$PGD | C-NH$_2$ | 0.19 | nd | 0.19 | nd | 0.25 | 0.06 |
| 11 | nC13CO[i] | GSSKSPSKKKPGD | C-NH$_2$ | 0.16 | >8 | 0.5 | nd | 0.09 | 0.125 |
| 12 | nC13CO[i] | GSSKSPSKKKP | C-NH$_2$ | 0.09 | >8 | 0.5 | nd | 0.06 | 0.007 |
| 13 | nC13CO[i] | GSKKK | C-NH$_2$ | 0.011 | 3 | 0.02 | nd | 0.0025 | 0.02 |
| 14 | nC13CO[i] | GSKKK | C-OH | <0.003 | 0.25 | <0.003 | nd | <0.003 | 0.09 |
| 15 | nC13CO[i] | KKK | C-OH | <0.003 | 0.19 | 0.02 | nd | <0.003 | 0.02 |
| *Lys-based linker* | | | | | | | | | |
| 16 | nC13CO[i] | GSKKK | K-OH | 0.06 | 0.5 | 0.007 | nd | <0.003 | nd |
| 17 | nC13CO[i] | KKK | K-OH | <0.003 | 0.06 | 0.03 | 0.05 | <0.003 | 0.125 |
| 18 | nC10CO[i] | KKK | K-OH | 0.045 | 0.093 | 0.5 | 0.5 | 0.06 | 1 |
| 19 | nC10CO[i] | KK | K-OH | 0.03 | 0.06 | 0.5 | 1 | 0.06 | 6 |
| 20 | nC10CO[i] | — | K-OH | 0.125 | 6 | 1.5 | nd | 0.19 | >8 |
| 21 | nC10CO[i] | KK | K-NH$_2$ | 0.007 | 0.06 | 0.06 | nd | 0.015 | 2 |
| 22 | nC10CO[i] | KK | K-NHMe | 0.015 | 0.06 | 0.125 | 0.5 | 0.015 | 4 |
| 23 | POB2K[k] | KK | K-OH | <0.003 | 0.25 | 0.5 | nd | <0.003 | 4 |
| 24 | POB2K[k] | KK | K-NHMe | <0.003 | 0.06 | 0.125 | 0.08 | <0.003 | 0.5 |
| 25 | Ac[l] | KK | K-OH | 4 | 4 | >8 | >8 | 4 | >8 |

MIC values are the median of a minimum of two independent determinations in duplicate
MIC minimum inhibitory concentration
[a]Membrane-insertive element
[b]Electrostatic effector peptide sequence
[c]Methicillin-resistant *S. aureus* ATCC 43300
[d]Human serum
[e]Vancomycin-intermediate *S. aureus* NRS1
[f]Vancomycin-resistant *S. aureus* NARSA VRS4
[g]Multi-drug-resistant *S. pneumoniae* ATCC 700677
[h]Vancomycin-resistant (VanA) *Enterococcus faecium* ATCC 51559
[i]nCxCO = *n*-alkanoyl
[j]4-Ph-Bz = 4-phenylbenzoyl
[k]POB2K = *N,N'*-bis(4-phenoxybenzoyl)-Lys
[l]Ac = acetyl

sequence did not compromise activity (**11–15**), and led to improved activity for a minimal basic KKK motif.

**Linker strategy leads to improvements in metabolic stability.** Initial in vivo studies indicated that the disulphide-linked conjugates possessed a very short half-life, consistent with in vitro studies showing instability in human plasma (Supplementary Table 2) or in the presence of physiological concentrations of glutathione (Supplementary Fig. 4). We therefore replaced the Cys-derived linker with a stable attachment through the side chain of Lys (Fig. 1, Y = CH$_2$, Supplementary Figs. 5–9), with the new compounds maintaining the desirable activity of the disulphide-linked pilot series. The best compounds were 20-fold to more than 100-fold more potent than vancomycin or daptomycin against MRSA, and significantly more active against VISA, VRSA and VRE (Tables 1 and 3), with inhibitory activity as good, or better than, recently approved glycopeptide antibiotics telavancin, dalbavancin and oritavancin. They showed similar high levels of potency (e.g. MIC ≤ 0.03 µg mL$^{-1}$) against other Gram-positive strains, including daptomycin-resistant MRSA, MDR *S. pneumoniae*, *S. pyogenes* and *S. epidermidis*, but were inactive against Gram-negative *Escherichia coli* (data not shown). Importantly, these new compounds were stable when exposed to plasma (Supplementary Table 2) and glutathione (Supplementary Fig. 4), and were resistant to microsomal metabolism (Supplementary Table 3).

**Vancapticins possess promising drug-like properties.** The lysine-linked compounds possessed low cytotoxicity against human cell lines (HepG2 and HEK293, with CC$_{50}$ over 1000-fold higher than the MIC for most compounds, even when tested with cells grown in only 1% serum to maximise cell sensitivity and availability of free compound) (Supplementary Table 4), and demonstrated minimal haemolysis of human erythrocytes in whole blood (most compounds were still non-haemolytic at 1600 µg mL$^{-1}$, over 10,000-fold higher than the MIC) (Supplementary Table 5). Antibiotic activity was measured both in the absence and presence of 50% human serum (Table 1), with the serum-induced reversal in activity (varying from 1-fold to 100-fold) used as a surrogate to estimate of the extent of protein binding. The compounds retained activity in the presence of an artificial lung surfactant (1% or 5% Survanta (Beractant), Supplementary Table 6), which reduces the activity of some antibiotics such as daptomycin[27] and makes them unsuitable for the treatment of lung infections. Vancapticin **24** was shown to be slowly bactericidal in a similar manner to vancomycin (Fig. 2).

**Table 2 Comparison of MIC activity of MIE-EEPS peptide vs. MIE-EEPS peptide-vancomycin conjugate vs. admixture of MIE-EEPS peptide plus vancomycin**

| Structure | | MIC (µg mL$^{-1}$) S. aureus MRSA ATCC 43300 | | | MIC (µg mL$^{-1}$) E. faecium MDR VanA ATCC 51559 | | |
|---|---|---|---|---|---|---|---|
| MIE-EEPS[a] peptide | Vanc conj. | MIE-EEPS peptide[b] | MIE-EEPS peptide/vanc mixture (1:1) | Vanc conj. | MIE-EEPS peptide[b] | MIE-EEPS peptide/vanc mixture (1:1) | Vanc conj. |
| Vanc | 1 | | 1 | | | >64 | |
| 8a[c] | 8 | >8 | 4 | 0.5 | >8 | 4 | 0.125 |
| 12a[d] | 12 | >8 | 4 | 0.09 | >8 | 4 | 0.007 |
| 13a[e] | 13 | >8 | 4 | 0.011 | >8 | 4 | 0.02 |
| 14a[f] | 14 | >8 | 4 | <0.003 | >8 | 4 | 0.09 |
| 15a[g] | 15 | >8 | 4 | <0.003 | >8 | 4 | 0.02 |

MIC measurements are the median of a minimum of two independent determinations in duplicate
[a]Membrane-insertive element—Electrostatic effector peptide sequence
[b]Note that the concentration of peptide on a molar basis is >twofold higher than when contained in the peptide-vancomycin derivative
[c]8a = nC13CO-GSSKSPSKKKKKKPGD-Cys-NH2
[d]12a = nC13CO-GSSKSPSKKKP-Cys-NH2
[e]13a = nC13CO-GSKKK-Cys-NH2
[f]14a = nC13CO-GSKKK-Cys-OH
[g]15a = nC13CO-KKK-Cys-OH

It is important that new antibiotics are not susceptible to innate or induced resistance. Compound **19** showed a low resistance frequency (a measure of the existing propensity for resistance) of $9.4 \times 10^{-10}$ against MRSA (ATCC 43300) (Supplementary Table 7). Both **17** and **24** induced low levels of resistance compared to daptomycin when MRSA (ATCC 43300) cultures were serially cultured over 20 days in the presence of increasing sub-lethal antibiotic concentrations (Fig. 3).

**Structure-activity relationships**. Structure-activity relationship studies using the stable linker series focused on variations in the insertive element and basic EEPS peptides. Subtle variations were observed between bacteria, with S. aureus and S. pneumoniae reaching maximal inhibition with dodecanoic acid (MIE = nC11CO), while E. faecium exhibited greater sensitivity toward longer alkyl groups >C12 (Table 3). However, these longer alkyl elements led to both increasing serum reversal and cytotoxicity, with undecanoic acid (MIE = nC10CO) providing an optimum compromise. In contrast, removal of the lipidic component altogether diminished activity (compare **25** vs. **19**) highlighting the importance of the insertive element for antimicrobial potency. Variation of the charged region from zero to three Lys residues (**18–20**) (Table 1) led to minor improvements in serum-free antibacterial activity, but the increasing charge caused variations in serum reversal and increased potency against E. faecium, particularly compared to **20** with no basic residue. A dibasic sequence (KK) was selected as optimum. Enantiomeric all D-amino acid effector sequences or linkers were equipotent with the natural L-amino acids, as were replacements of Lys with Orn or Dab, suggesting that no chiral molecular recognition was involved (Table 3).

**Demonstration of efficacy against MRSA and MDR S. pneumoniae**. We next examined the PK profiles of compounds **18**, **19**, **21** and **24** in mice (Fig. 4). When compared to the murine profiles of daptomycin[28] and telavancin[29], our compounds exhibited profiles consistent with extrapolated once-daily dosing in man, representing a significant advantage over the twice daily dosing most commonly used for vancomycin. In mouse plasma, compound levels were maintained at concentrations above the MRSA MIC (determined in the presence of serum) for over 12 h following a single 10 mg kg$^{-1}$ subcutaneous (SC) dose (Fig. 4, Supplementary Tables 8 and 9). While the PK/PD driver for vancomycin is now believed to be the area under the unbound drug concentration-time curve [$f$AUC]/MIC, historically the percentage of a 24-h time period that the unbound drug concentration exceeds the MIC [$fT_{>MIC}$] was initially reported to be a better predictor of the antibacterial effect so is still useful as an approximation[30].

The new compounds showed potent bactericidal efficacy in a neutropenic mouse thigh infection model against MRSA. A single 25 mg kg$^{-1}$ subcutaneous (SC) dose of **19** was equivalent to a single 200 mg kg$^{-1}$ dose of vancomycin, with a 6-log reduction compared to the saline control at 24 h (Fig. 5a). Activity was still maintained at 5 mg kg$^{-1}$ for **19**, while vancomycin was almost inactive at 25 mg kg$^{-1}$. Highly protein bound **24** was less effective, but other analogues such as **21** were equally active. Both **19** and **24**, when tested in an alternate lung infection model using intratracheal administration of a lethal dose of S. pneumoniae, resulted in 100% survival after 10 days (Fig. 5b). The results from the pneumonia model were consistent with in vitro MIC testing that showed activity was maintained in the presence of artificial lung surfactant, in contrast to the loss of activity observed with daptomycin, which is ineffective at treating pneumonia[31]. Finally, **19** and **24** were tested in an intra-peritoneal (IP) infection model using bioluminescent methicillin-sensitive S. aureus (MSSA) Xen-29, with imaging at 9 h post infection again showing good efficacy for **19**, but a reduced activity for **24** (Fig. 5c–e).

**Assessment of Lipid II binding mechanism of action**. Vancomycin acts by inhibiting the transpeptidase step of peptidoglycan synthesis, although there is also evidence of blocking the previous transglycosylase step[32]. Kahne and co-workers have reported that vancomycin group antibiotics with lipophilic substituents (e.g. a biphenyl group) differ from vancomycin by acting primarily via transglycosylase inhibition[33]. The remarkable activity of our compounds could be due to a number of factors: (i) an increased affinity for the target terminal D-Ala-D-Ala dipeptide of Lipid II, (ii) enhanced dimerisation leading to higher avidity for membrane-bound Lipid II[34], (iii) an additional or different mode of action or target than vancomycin (such as membrane permeabilisation), and/or (iv) membrane localisation resulting in an enhanced drug concentration at the membrane surface, which could serve to enhance any of the above.

Compounds **17** and **24** inhibited peptidoglycan formation in a cell free radiolabelled synthesis assay that monitors the incorporation of $^{14}$C-labelled N-acetylglucosamine [$^{14}$C-GlcNAc] into Lipid II and then into peptidoglycan, with Lipid II accumulating at tenfold lower concentration of compound compared to vancomycin (Figs. 6a-c). A competitive ligand

**Table 3 Structure activity relationships of carbon-linked vancapticins**

| | Structure | | | MIC (µg mL$^{-1}$) | | | | | | |
| Id | MIE | EEPS | Linker | MRSA$^a$ | MSSA$^b$ | +50% HS$^c$ | VISA$^d$ | VRSA$^e$ | MDR SP$^f$ | VRE$^g$ |
|---|---|---|---|---|---|---|---|---|---|---|
| 1 | | Vancomycin | | 1 | 1 | 1 | 4 | >64 | 1 | >64 |
| 2 | | Telavancin | | 0.06 | nd | 0.5 | 0.25 | 1 | 0.06 | 2 |
| 3 | | Dalbavancin | | 0.03 | ≤0.016 | 8 | 1 | 16 | 0.09 | >8 |
| 4 | | Daptomycin | | 2 | 1 | 3 | 12 | 3 | 2 | 16 |
| 5 | | Oritavancin | | 0.06 | ≤0.016 | 0.5 | 1 | 0.125 | 0.06 | 0.125 |
| *Effect of chirality—EEPS and linker* | | | | | | | | | | |
| 24 | POB2K$^h$ | KK | K-NHMe | <0.003 | ≤0.016 | 0.06 | 0.125 | 0.08 | <0.003 | 0.5 |
| 25 | POB2K$^h$ | kk | K-NHMe | 0.0007 | nd | 0.5 | 0.125 | nd | 0.015 | 2 |
| 26 | POB2K$^h$ | kK | K-NHMe | <0.003 | nd | 0.25 | 0.19 | nd | <0.003 | 1.5 |
| 27 | POB2K$^h$ | Kk | K-NHMe | 0.03 | nd | 0.25 | 0.75 | nd | 0.04 | 10 |
| 28 | POB2K$^h$ | KK | k-NHMe | <0.003 | nd | 0.06 | 0.125 | nd | 0.015 | 0.75 |
| *Effect of side chain length—EEPS and linker* | | | | | | | | | | |
| 24 | POB2K$^h$ | KK | K-NHMe | <0.003 | nd | 0.06 | 0.125 | 0.08 | <0.003 | 0.5 |
| 27 | POB2K$^h$ | K-Dap | K-NHMe | 0.0045 | nd | 0.375 | 0.0925 | nd | 0.016 | 1.5 |
| 28 | POB2K$^h$ | Dap-K | K-NHMe | 0.0045 | nd | 0.25 | 0.155 | nd | 0.011 | 0.75 |
| 29 | POB2K$^h$ | Dap-Dap | K-NHMe | 0.0045 | nd | 0.5 | 0.0925 | nd | 0.0085 | 0.75 |
| 30 | POB2K$^h$ | K-Dab | K-NHMe | <0.003 | nd | 0.06 | 0.0925 | nd | <0.003 | 0.75 |
| 31 | POB2K$^h$ | Dab-K | K-NHMe | <0.003 | nd | 0.045 | 0.0925 | nd | <0.003 | 0.375 |
| 32 | POB2K$^h$ | K-Orn | K-NHMe | <0.003 | nd | 0.19 | 0.0375 | nd | <0.003 | 1 |
| 33 | POB2K$^h$ | Orn-K | K-NHMe | <0.003 | nd | 0.125 | 0.125 | nd | 0.007 | 1.5 |
| 34 | POB2K$^h$ | Orn-Orn | K-NHMe | <0.003 | nd | 0.25 | 0.0185 | nd | <0.003 | 0.5 |
| 35 | POB2K$^h$ | KK | Orn-NHMe | <0.003 | nd | 0.16 | 0.25 | nd | 0.005 | 1.25 |
| *Effect of MIE on activity* | | | | | | | | | | |
| 36 | nC9CO-$^i$ | KKK | K-OH | 0.06 | nd | 0.06 | 0.75 | 0.75 | 0.06 | 8 |
| 18 | nC10CO- | KKK | K-OH | 0.045 | nd | 0.093 | 0.5 | 0.5 | 0.06 | 1.03 |
| 37 | nC11CO- | KKK | K-OH | 0.005 | nd | 0.03 | 0.06 | 0.75 | <0.003 | 0.5 |
| 38 | nC12CO- | KKK | K-OH | <0.003 | nd | 0.045 | 0.03 | nd | <0.003 | 0.1875 |
| 17 | nC13CO- | KKK | K-OH | <0.003 | ≤0.016 | 0.06 | 0.03 | 0.045 | <0.003 | 0.125 |

$^a$Methicillin-resistant *S. aureus* ATCC 43300
$^b$Methicillin-sensitive S. aureus ATCC 29213
$^c$Human serum
$^d$Vancomycin intermediate *S. aureus* NRS1
$^e$Vancomycin-resistant *S. aureus* NARSA VRS4
$^f$Multidrug-resistant *S. pneumoniae* ATCC 700677
$^g$Vancomycin-resistant (VanA) *Enterococcus faecium* ATCC 51559
$^h$POB2K N,N'-bis(4-phenoxybenzoyl)-Lys
$^i$nCxCO n-alkanoyl, *nd* not determined. MIC measurements are the median of a minimum of two independent determinations in duplicate

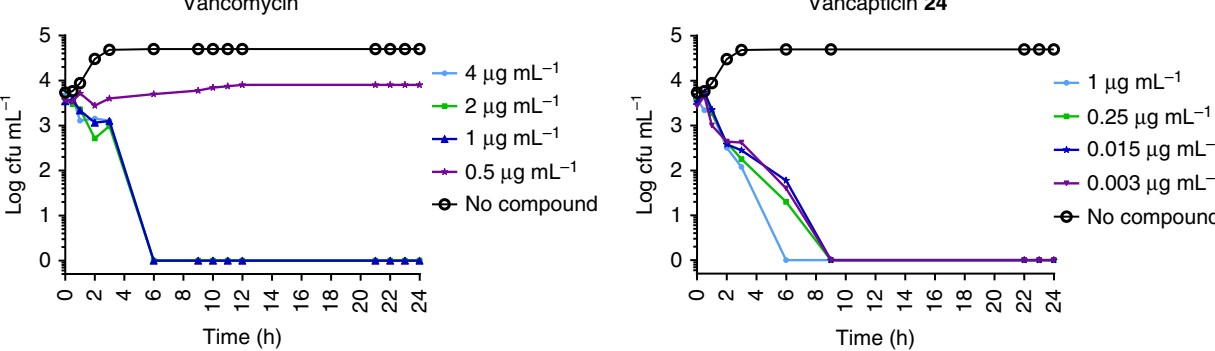

**Fig. 2** Time kill assay. Bactericidal activity of vancomycin **1** and vancapticin **24** against *S. aureus* ATCC 43300 as measured by agar plate dilution colony measurement of wells from broth microdilution MIC measurement over time. Data are $n = 1$

antagonism assay (Fig. 6e), in which the MIC was tested in the presence of a Lipid II binding site competitor, Ac-Lys(Ac)-D-Ala-D-Ala (Ac$_2$Kaa), required a 100-fold molar excess of Ac$_2$Kaa to ablate the activity of vancomycin. In contrast, the new compounds required over 10,000-fold molar excess to neutralise antibiotic action, also confirming that the new analogues still

targeted Lipid II and peptidoglycan synthesis (Fig. 6e, Supplementary Table 10). Isothermal titration calorimetry (ITC) assessing glycopeptide binding to the C-terminal tri-peptide or penta-peptide components of Lipid II showed overall similar binding as vancomycin, with increased enthalpy offset by increased entropy (Fig. 6d, Supplementary Fig. 11). The

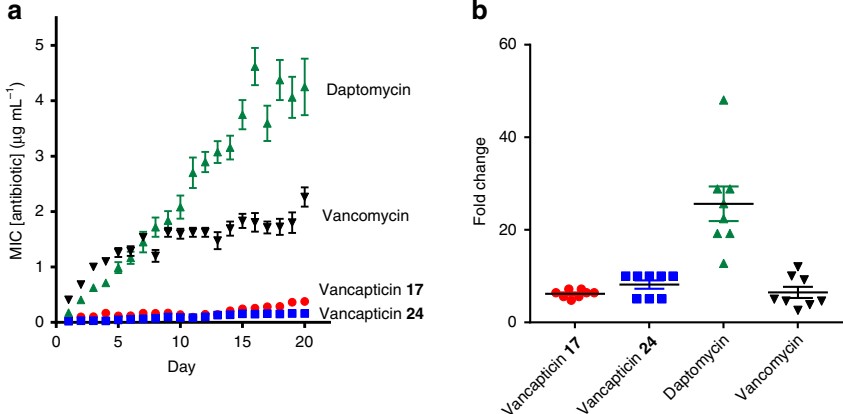

**Fig. 3** Resistance induction. **a** Average daily MIC for MRSA (ATCC 43300) grown with increasing sub-lethal concentrations of vancomycin, daptomycin, compound **17** and compound **24** over 20 days of bacterial growth. **b** The corresponding overall fold-increase in MRSA MIC for the four compounds. Data are mean ± SEM for $n = 8$

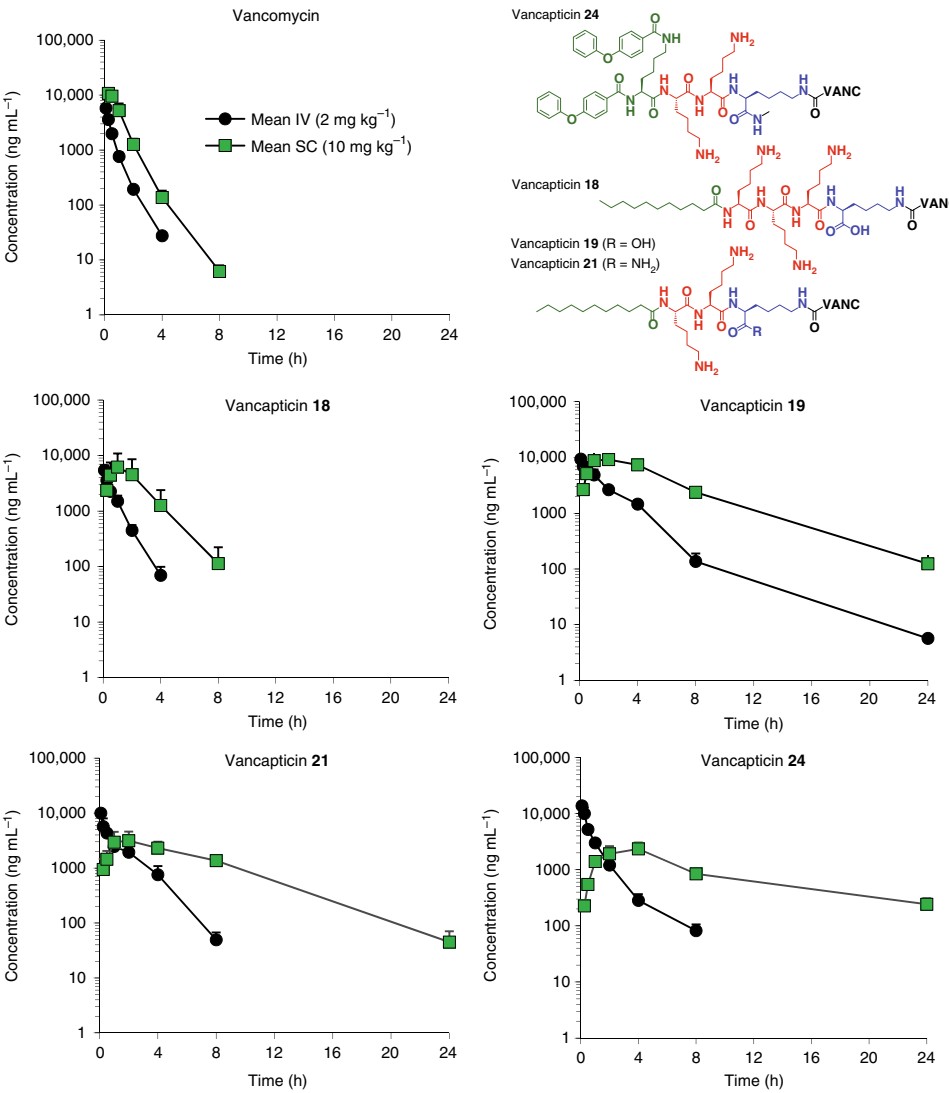

**Fig. 4** Mouse PK. Pharmacokinetic profiles with plasma concentrations of vancomycin and compounds **18**, **19**, **21** and **24** dosed at 2 mg kg$^{-1}$ intravenously (IV) or 10 mg kg$^{-1}$ subcutaneously (SC) in mice; serial sampling. Data are mean + S.D. for $n = 3$. See Supplementary Tables 8 and 9 for individual mouse data and calculated parameters

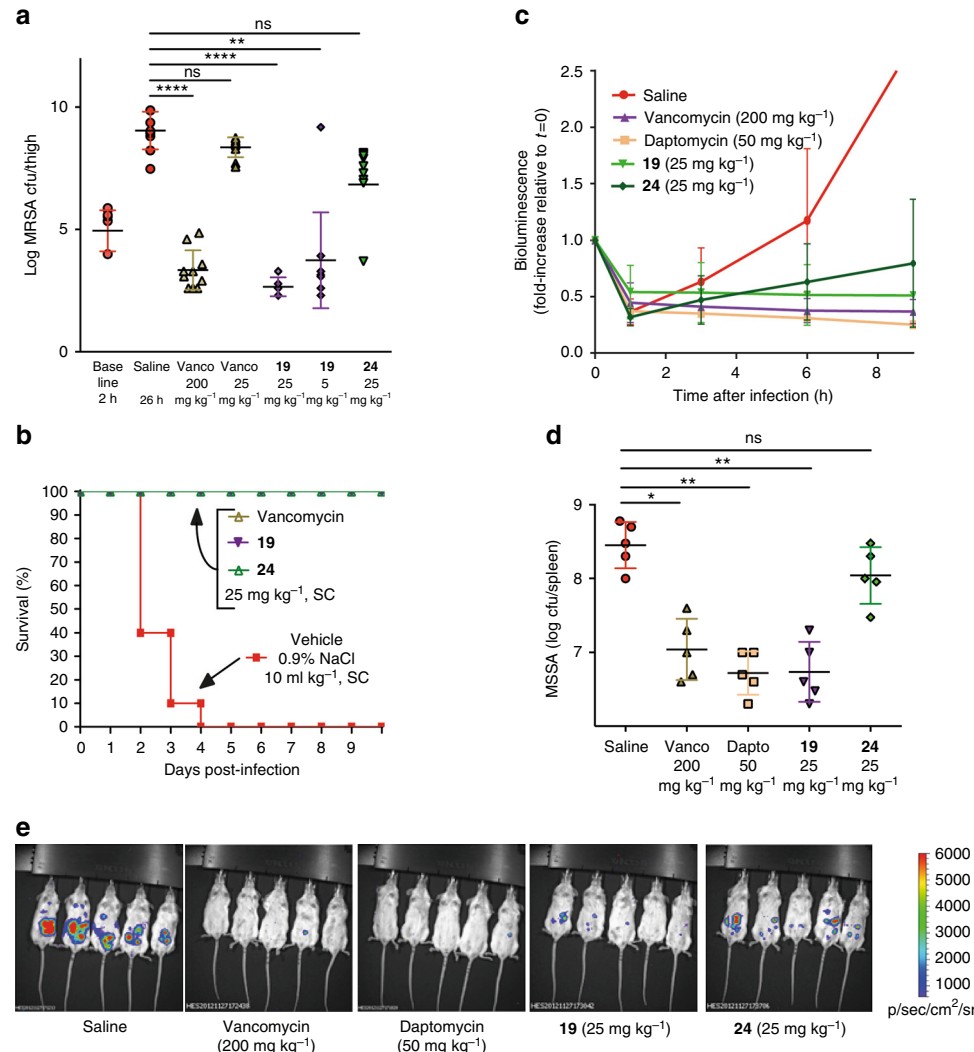

**Fig. 5** Mouse efficacy models. **a** MRSA thigh infection model. Colony forming units (CFU) in neutropenic mice infected in each thigh with MRSA, followed 2 h later by a single subcutaneous dose of antibiotic at concentrations indicated, with sacrifice and bacterial load determination in homogenised thighs at 24 h post treatment (26 h post infection): $n = 10$ (five mice per group, two thighs per mouse; errors are mean ± SEM). A significant difference was found between saline at 26 h and vancomycin **1** at 200 mg kg$^{-1}$ ($p < 0.0001$), and vancapticin **19** at both 25 mg kg$^{-1}$ ($p < 0.0001$) and 5 mg kg$^{-1}$ ($p = 0.0011$). No statistical difference was found for vancomycin **1** at 25 mg kg$^{-1}$ or vancapticin **24** at 25 mg kg$^{-1}$ compared to saline. Vancapticin **19** at 25 mg kg$^{-1}$ also showed a significant reduction compared to the initial $t = 2$ h baseline inoculum ($p = 0.0014$) whereas vancomycin **1** at 200 mg kg$^{-1}$ was not statistically significant. Statistical analysis done using Graph Pad, 1-way ANOVA, Bonferroni post-test. **b** S. pneumoniae lung infection LD$_{90}$. Survival of healthy mice infected with a lethal intratracheal dose of S. pneumoniae ATCC6301 followed 2 h later by a single subcutaneous dose of antibiotic at 25 mg kg$^{-1}$ ($n = 10$ mice per group). **c–e** Bioluminescent MSSA intraperitoneal model. Neutropenic mice were injected intraperitoneally with $2.5 \times 10^7$ CFU bioluminescent MSSA Xen-29 (possessing a stable copy of the Photorhabdus luminescens lux operon on the bacterial chromosome) then treated after 0.5 h with subcutaneous doses of saline, 200 mg kg$^{-1}$ vancomycin, 50 mg kg$^{-1}$ daptomycin, 25 mg kg$^{-1}$ **19** or 25 mg kg$^{-1}$ **24** ($n = 5$ mice per group). **c** Changes in total flux levels. Variations in bioluminescence were measured at $T = 0, 1, 3, 6$ and 9 h, quantified with the IVIS Living Image software where the total flux (number of photons/second) was calculated by a user defined region of interest (ROI) covering the infection sites. The saline treated group showed a large increase in the total flux, particularly after 3 h ($T = 3$ h). Daptomycin, vancomycin and **19** treated groups showed a progressive reduction in the bioluminescence signal 1 h after inoculation, while the signal detected from **24** increased slightly. All antibiotic administered groups showed reduced bioluminescence signal at $T = 9$ h compared to immediately after inoculation ($p < 0.001$ for all groups; errors are mean ± S.D.). **d** Changes in CFU per spleen after 9 h. Individual spleens were homogenised and diluted for plating. The calculated CFU/spleen counts for five mice are presented, along with the mean (black bar); errors are mean ± S.D. A significant difference was found between saline and vancomycin **1**, daptomycin **4** and compound **19** ($p < 0.001$ for all groups). No statistical difference was found for **24** compared to saline, while this compound was found statistically less efficacious than **1** ($p < 0.01$), **4** ($p < 0.001$), and **19** ($p < 0.001$). Statistical analysis done using Graph Pad, 1-way ANOVA, Bonferroni post-test. **e** Bioluminescent images at $T = 9$ h. Bioluminescent imaging of infected mice at $T = 9$ h using Xenogen IVIS-200 Optical In Vivo Imaging System (PerkinElmer)

propensity of compounds to dimerise (also determined by ITC) was found to be similar to that for vancomycin (Supplementary Table 11, Supplementary Fig. 12), and increased in the presence of ligand. Hence, neither enhanced ligand binding nor dimerisation was responsible for the improved potency.

**Assessment of membrane binding and disruption mechanism of action**. Increased membrane association of the new compounds **17** and **24** compared to vancomycin was supported by surface plasmon resonance (SPR) studies. There was a propensity to bind more strongly to anionic (dimyristoyl-L-α-phosphatidyl-

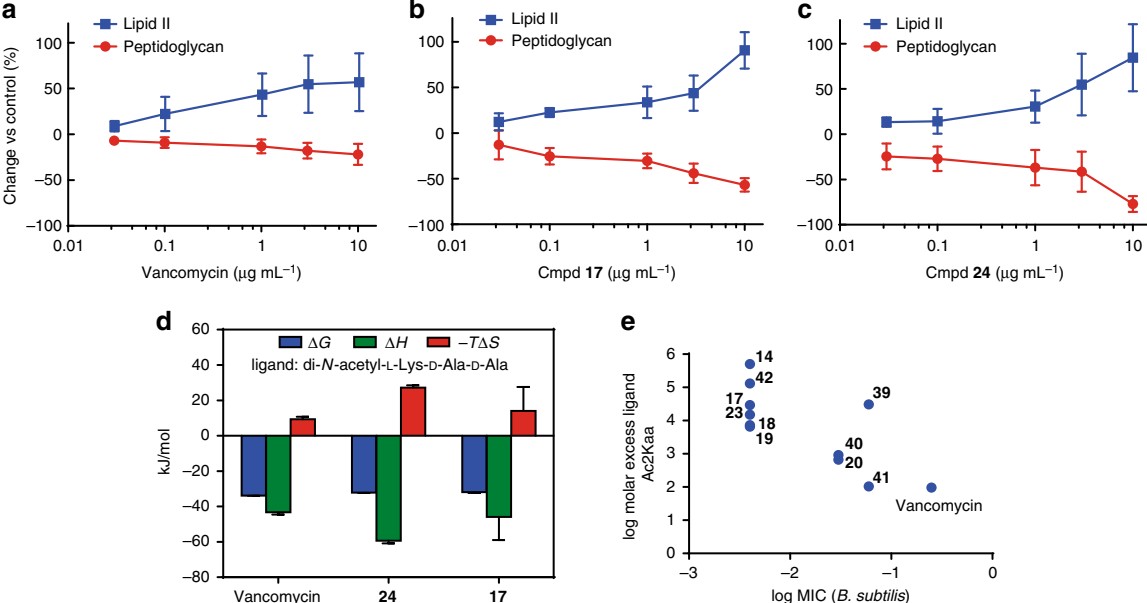

**Fig. 6** Lipid II binding effects. **a–c** Accumulation of Lipid II (blue) and % inhibition of peptidoglycan synthesis (red) measured in cell-free assay system using [14]C-labelled Lipid II precursor as a function of antibiotic concentration (errors are mean ± S.D, $n = 3$) for (**a**) vancomycin, (**b**) Cmpd **17** and (**c**) Cmpd **24**. **d** ITC data for binding to Ac-Lys(Ac)-D-Ala-D-Ala (errors are mean ± S.D, $n = 3$). **e** molar excess of ligand Ac-Lys(Ac)-D-Ala-D-Ala (KDADA) required to antagonise antibacterial activity of compounds against *B. subtilis* (ATCC 6633) compared to the MIC of compounds

L-glycerol, DMPG) over zwitterionic (dimyristoyl-L-α-phosphatidylcholine, DMPC) lipid bilayers (Fig. 7a). To discount the possibility that enhanced antimicrobial activity resulted from non-specific membrane-based perturbation effects similar to those induced by cationic antimicrobial peptides[35], membrane effects of the new compounds were assessed. Aggregation was discounted as a source of membrane activity as the critical micelle concentration (CMC) was found to exceed 100 μM for selected compounds **17**, **19** and **23**. Furthermore, compounds were generally non-haemolytic against human erythrocytes in whole blood at 1600 μg mL⁻¹, with any haemolysis observed not correlating with lipophilicity (Fig. 7b and Supplementary Table 5).

Evidence for membrane perturbation was obtained in the presence of membrane-selective fluorescent probes. First, we employed the membrane potential-sensitive cationic fluorescent probe 3,3′-dipropylthiadicarbocyanine iodide [diSC3(5)], which is prevented from partitioning to the surface of polarised cells during disruption of membrane potential, causing dye release into the media[36]. Here, analogues **17** and **24** were observed to dissipate the membrane potential of MRSA (ATCC 43300) to a similar or greater degree than the control antibiotic oritavancin **5**[37, 38], as measured by the fluorescence output of diSC3(5) (Fig. 7c). All compounds were dosed at 16 μg mL⁻¹, equating to a concentration approximately 16-fold above their MIC for **5**, **17** and **24** (Note that membrane permeabilisation studies were conducted in polystyrene plates without polysorbate-80 additive to avoid assay interferences; under these conditions the MIC for **5**, **17** and **24** = 1–2 μg mL⁻¹—Supplementary Table 12). The effect was less pronounced for compound **19** despite a higher MIC-fold exposure (~128-fold MIC), whereas vancomycin, daptomycin and dalbavancin showed no effect under the conditions of the assay. Similar effects were observed against MSSA (ATCC 29213) (Supplementary Fig. 13).

Additional evidence for membrane perturbation was obtained with the membrane impermeable dye propidium iodide (PI), which stains nuclear chromatin upon cell membrane disruption, resulting in fluorescence enhancement[39]. To this end, MRSA (ATCC 43300) was incubated with selected antibiotic compounds

at 16 μg mL⁻¹ for 1 h (~16-fold MIC for compounds **17**, **24**, oritavancin, dalbavancin and vancomycin in polystyrene plates) followed by exposure to propidium iodide (Fig. 7d). Again, compounds **17** and **24** elicited a response with comparable fluorescence intensity to oritavancin[37, 38]. Daptomycin[40] caused approximately 4–6-fold higher fluorescence than oritavancin or **17**. In contrast, compound **19**, vancomycin and dalbavancin showed little to no effect in this membrane permeabilisation assay. A similar trend was observed against MSSA (ATCC 29213) (Supplementary Fig. 14).

## Discussion

Whilst there is a current perception that Gram-positive antibiotics are less urgently required than Gram-negative therapies, in reality the number of drug-resistant Gram-positive infections, and resulting deaths, far outnumber those from resistant Gram-negative bacteria[1]. Over the past four decades the rational improvement of existing classes of antibiotics has proven to be a very successful strategy. Vancomycin is a logical candidate for improvement via specific membrane-targeting, given its mode of action and the length of time it took (more than 20 years) for significant clinical resistance to arise.

Membrane-ligand interactions are an underexplored area of drug design. We created a series of membrane-targeted vancomycin derivatives with significantly improved activity against a range of Gram-positive bacteria, including vancomycin-resistant strains. The modular nature of their structure (Fig. 1), with a lipophilic group (MIE) designed to insert in the membrane, a positively charged peptide (EEPS) for electrostatic interaction with the overall negative surface charge of bacterial membranes, and a linker for attachment to the C-terminal carboxyl group of vancomycin, provided a platform to readily produce multiple derivatives using solid phase peptide synthesis. A series of structure-activity relationship studies resulted in promising compounds with potent in vitro activity, often more than 100-fold more active than vancomycin (Tables 1 and 3). Both MIE and EEPS components were required for enhanced activity.

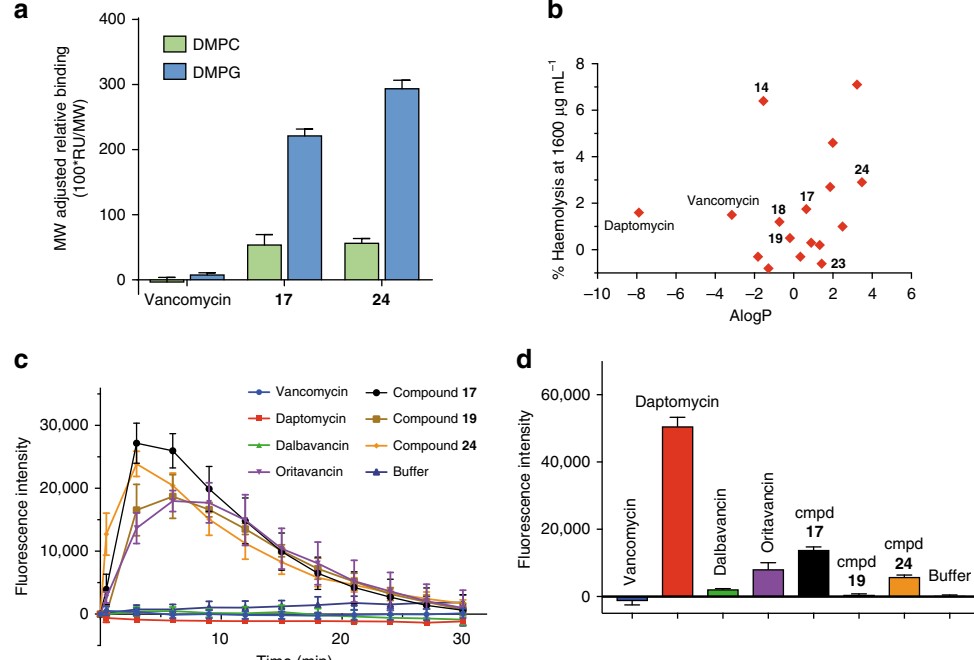

**Fig. 7** Membrane effects. **a** SPR binding of vancomycin, **17** and **24** to L1 chips coated with lipid bilayers formed from DMPC or DMPG (errors are mean ± S.D, n = 3). **b** % haemolysis of human red blood cells in whole blood induced by a set of derivatives at 1600 µg mL$^{-1}$ relative to complete lysis, compared with their calculated lipophilicity (n = 2). **c** Membrane depolarisation: change in fluorescence intensity of the reporter dye diSC3(5) over 30 min following treatment of *S. aureus* ATCC 43300 (early exponential phase) with test compounds at 16 µg mL$^{-1}$ (errors are mean ± S.D, n = 4). **d** Membrane permeabilisation: maximal fluorescence intensity of the reporter dye propidium iodide after 2 h following incubation of *S. aureus* ATCC 43300 (early exponential phase) with test compounds (16 µg mL$^{-1}$) for 1 h at 37 °C (errors are mean ± S.D, n = 4)

Converting the linker from a disulphide bridge to a carbon-based chain substantially improved the drug-like properties of the constructs, restoring plasma stability. Selected candidates demonstrated excellent in vivo activity in multiple murine models of infection at doses up to 40-fold lower than vancomycin (Fig. 5), with a pharmacokinetic (PK) profile consistent with once daily dosing in humans (Fig. 4). Induced resistance experiments demonstrated that the compounds retained a low propensity to cause resistance, in contrast to daptomycin (Fig. 3).

The remarkably potent activity possessed by these antibiotics is most likely due to a membrane anchoring chelate effect that enhances antibiotic binding to cell wall precursors at the cell surface. In effect, intra-molecular binding is achieved as both the antibiotic and its target are attached via membrane anchors to the same template. Examining peptidoglycan synthesis in a cell-free system demonstrated that the vancapticins caused greater inhibition of biosynthetic activity than vancomycin, corresponding with much greater quantities of Ac-Lys(Ac)-D-Ala-D-Ala ligand in free solution required to antagonise vancapticin antimicrobial activity against live bacteria (Fig. 6).

The enhanced ability to inhibit peptidoglycan synthesis is accompanied by membrane binding and disruption (Fig. 7). SPR studies demonstrated a greater propensity for binding to an anionic lipid bilayer compared to vancomycin. A diSC3(5) dye-based assay in live bacteria showed that vancapticins induced membrane depolarisation to a similar extent to that seen with oritavancin, with minimal activity observed with vancomycin or dalbavancin. In contrast, a membrane permeabilisation assay (propidium iodide dye) gave variable results for the same vancapticins, with **17** and **24** eliciting a comparable response as oritavancin, but fourfold to sixfold less than daptomycin. Vancapticin **19** and dalbavancin were much less active, while vancomycin had no effect. The increased membrane activity of the vancapticins appears to enhance their potency, but does not

translate into increased killing kinetics: compound **24** shared a similar slowly bactericidal profile with vancomycin (Fig. 2).

In summary, we report the design and development of promising new semisynthetic lipoglycopeptides intended to selectively interact with bacterial membranes. Further improvements and additional studies are ongoing to assess the potential to advance the vancapticins as a new clinical candidate for the treatment of drug-resistant Gram-positive infections.

## Methods

**Ethics approvals.** Research was conducted under an Institutional Animal Care and Use Committee approved protocol in compliance with the Animal Welfare Act, PHS Policy and other federal statutes and regulations relating to animals and experiments involving animals. Contract facilities where animal studies were conducted were accredited by the Association for Assessment and Accreditation of Laboratory Animal Care, International and adhere to principles stated in the Guide for the Care and Use of Laboratory Animals, National Research Council, 2011. Studies performed at the University of Queensland were approved by the Molecular Biosciences Animal Ethics Committee. Sample size for in vivo studies was selected based on minimising animal use while providing sufficient data points to show significant differences based on historical studies. No animals were excluded from analysis. No documented method of randomisation was employed, with animals randomly assigned to groups. Code numbers were employed for all in vivo studies, so the study investigator was blinded to the actual identity of compounds. Human ethics approval from the University of Queensland Medical Research Ethics Committee was obtained for use of human blood for haemolysis studies.

**Chemical synthesis. General:** C-terminal amide or methyl amide containing peptides were assembled on Fmoc-Rink Amide AM polystyrene resin (Iris Biotech GmbH) or HypoGel hydroxymethylbenzoic acid (HMBA) resin (Iris Biotech GmbH), respectively. Peptides bearing free C-terminal carboxylic acids were assembled on Fmoc-Cys(Trt) or Fmoc-Lys(Boc) Wang polystyrene resin. 9-Fluorenylmethyloxycarbonyl (Fmoc)-L-amino acids utilised standard side-chain protecting groups: *tert*-Butyl (Asp, Ser), Trityl (Cys), *tert*-Butoxycarbony (Boc) or 4-methyltrityl (Mtt) or 1-(4,4-dimethyl-2,6-dioxocyclohexylidene)-3-methylbutyl (ivDde) (Lys, Dab, Dap, Orn) and were purchased from Iris Biotech GmbH, as were the coupling reagents *O*-(benzotriazol-1-yl)-*N,N,N′,N′*-tetramethyluronium hexafluorophosphate (HBTU) and *O*-(benzotriazol-1-yloxy)tripyrrolidinophosphonium hexafluorophosphate (PyBop). Azabenzotriazol-1-yl-*N,N,N′,N′*-

tetramethyluronium hexafluorophosphate (HATU) was sourced from GenScript Corporation (USA), and (S)-pyridylthio cysteamine.HCl from Toronto Research Chemicals Inc. Vancomycin.HCl **1** was purchased from Hallochem Pharma Co. Ltd, daptomycin **4** from Molekula (Cat # 64342447) and oritavancin diphosphate **5** (Cat # 317101), and dalbavancin.HCl **3** (Cat # 317136) from MedKoo Biosciences. Dalbavancin **3** was also synthesised internally, as was telavancin **2**. Analytical liquid chromatography / mass spectrometry (LC/MS) was performed using reverse-phase high-performance liquid chromatograph (RP-HPLC) on an Eclipse XDB-Phenyl column (5 μm, 4.6 × 150 mm, 1 mL min⁻¹) coupled to an Agilent 1200 series system with detection by a single quadrupole mass spectrometer (6110), a UV detector operating at 210 nm and an evaporative light scattering detector (ELSD). Preparative HPLC was performed using reverse-phase HPLC on an Eclipse XDB-Phenyl column (5 μm, 21.2 × 100 mm, 20 mL min⁻¹) coupled to an Agilent 1260 Infinity system with detection by UV (210 nm). Elution was effected using appropriate gradients with 0.05% formic acid (analytical) or 0.1% trifluoroacetic acid (TFA) (preparative) in water (solvent A) and 0.05% formic acid (analytical) or 0.1% TFA (preparative) in acetonitrile (solvent B). (+)-electrospray ionisation mass spectrometry/mass spectrometry (ESI–MS/MS) were obtained using an API QSTAR™ Pulsar Hybrid LC-MS/MS System using the following conditions: column oven 40 °C, MS parameters: CUR: 20, IS: 5300, TEM: 450, GS1: 45, GS2: 45, ihe: ON, DP 60. HPLC Column: Waters Atlantis® T3 5 μm 2.1 × 50 mm with Atlantis® T3 5 μm 2.1 × 10 mm guard column. Flow rates and solvent: 0.4 mL min⁻¹, solvent A: 0.1% formic acid in H₂O, solvent B: 0.1% formic acid in acetonitrile; isocratic 0% B from 0→1 min, gradient 0%→100% B from 2→9 min, isocratic 100% from 5→12 min, gradient 100%→0% B from 12→12.5 min and isocratic 2% B from 12.5→17.5 min. High resolution mass spectrometry (HRMS) was performed on a Bruker Micro TOF time-of-flight (TOF) mass spectrometer using (+)-ESI calibrated to NH₄OAc.¹H (600 MHz), ¹³C (150 MHz) and 2D NMR spectra were obtained in d₆-dimethylsulphoxide (DMSO) calibrated to TMS resonances (δ_H 0.00, δ_C 0.0) using a Bruker Avance-600 spectrometer equipped with a TXI cryoprobe.

### General procedure A: Loading of Fmoc-L-Lys(ivDde)-OH or Fmoc-L-Orn (ivDde)-OH onto hypogel HMBA resin.
HypoGel HMBA resin was washed with anhydrous dimethylformamide (DMF) (3×) under an inert atmosphere. A mixture of Fmoc-L-Lys(ivDde)-OH or Fmoc-L-Orn(ivDde)-OH (5 eq), hydroxybenzotriazole (5 eq) in anhydrous DMF (2.5 mL per 1 mmol amino acid) was treated with 1,3-diisopropylcarbodiimide (DIC) (5 eq) and 4-dimethylaminopyridine (0.3 eq). The resulting solution was added to the resin and the resin was shaken at room temperature overnight. The resin was drained, and successively washed with DMF, methanol (MeOH) and dichloromethane (DCM) (3 × each). A solution of acetic anhydride (4 eq) and diisopropylethylamine (DIPEA) (8 eq) in DMF (~10 mL per 1 gram of resin) was added to the resin to block any unreacted sites. After shaking for 1 h, the resin was drained and washed with DMF, MeOH and DCM (3 × each) and dried in vacuo.

### General procedure B: Standard Fmoc solid phase peptide synthesis.
The starting resin was washed with anhydrous DMF (3 ×). Fmoc-deprotection was performed by shaking the resin with piperidine in DMF (20:80 v/v, 10 mL per 1 g of resin) for 30 min. The resin was drained and washed successively with DMF, MeOH, DCM and DMF (3 × each). A solution of Fmoc-protected amino acid or fatty acid (4 equiv) in DMF (~0.4 M) was treated with HBTU (4 equiv) and DIPEA (16 equiv). After 10 min, the activated amino acid solution was added to the resin. The resin was agitated at room temperature (RT) for 1 h, drained, washed successively with DMF, MeOH and DCM (3 × each), and dried in vacuo. Coupling efficiency was assessed using the Kaiser test and/or by cleavage of 2–3 mg of resin and analysis by LCMS. If required, the coupling reaction was repeated.

### General procedure C: Removal of the ivDde protecting group.
Crude lipopeptide was treated with an excess of hydrazine hydrate (2% v/v in DMF, ca. 10 eq) at RT. Reaction progress was monitored by LC/MS. Upon completion, the solution was concentrated in vacuo and the resulting crude product was lyophilised (ACN/H₂O). The material was then used in the next step without further purification (intermediate) or purified by preparative HPLC (final product).

### General procedure D: Removal of the 4-methyltrityl (Mtt) group.
The Mtt-protected vancomycin derivative was treated with a solution of TFA/triethylsilane (TES)/DCM (2:5:93, 40 mL per 1 g of crude peptide) at room temperature for 1 h and concentrated under reduced pressure. The process was repeated if deprotection was incomplete as determined by LCMS analysis. Upon completion, the solution was concentrated in vacuo and the crude product was purified by preparative HPLC.

### General procedure E: Resin cleavage.
(i) Rink-amide, Wang- and 2-CTC-resins. Peptides incorporating cysteine were cleaved from the resin by agitation with TFA/1,2-ethanedithiol (EDT)/triisopropylsilane (TIPS)/water (90/5/2.5/2.5 v/v/v/v, 20 mL per 1 g of resin). Non-cysteine containing peptides were cleaved from the resin by agitation with TFA/TIPS/water (95:2.5:2.5 v/v/v, 20 mL per 1 g of resin). Resins were filtered and washed with TFA (3×). The combined filtrates were concentrated in vacuo and the crude peptides were precipitated with cold diethyl ether, isolated

by centrifugation and lyophilised (ACN/H₂O). (ii) HMBA hypogel resin: C-terminal N-methyl amides. Prior to cleavage, the resin was washed with tetrahydrofuran (THF) (3×) and DIPEA (2×) and then treated with methylamine (2 M in THF, 20 mL per 1 g of resin) with overnight agitation at RT. The resin was drained, and washed with THF, DCM and ACN (3× each). The combined filtrates were removed under reduced pressure to afford crude product.

### eneral procedure F: Disulphide exchange.
GPeptides incorporating cysteine were dissolved in acetonitrile/water (1:1, ca. 150 mL per 1 mmol of substrate), and the pH was adjusted (pH 8–9) by addition of DIPEA (ca. 5 eq). A solution of **43** (1.5 equiv) in acetonitrile/water (1:1, ca. 100 mL per 1 mmol of **43**) was added. After standing at room temperature for 1 h, the reaction mixture was concentrated by lyophilisation, and the resulting crude product was purified by preparative RP-HPLC.

### General procedure G: Solution phase coupling with vancomycin.
A stirred solution of peptide (1 eq), O'-(7-Azabenzotriazol-1-yl)-N,N,N',N'-tetra-methyluronium hexafluorophosphate (HATU) (1.3 eq) and vancomycin hydrochloride (1.2 eq) in anhydrous DMF (~ 30 mL per 1 mmol of peptide) was cooled to 0 °C. After 30 min, DIPEA was added (10 eq), and stirring was continued for 1 h at room temperature. The solvent was evaporated in vacuo and the residue was lyophilised from ACN/H₂O. The crude product was purified by preparative RP-HPLC.

### Structure elucidation of synthesised compounds.
Compound purity was determined using LC-(+)-ESI-MS/UV/ELSD and their molecular formulae determined by (+)-HR-ESI-MS. (+)-ESI-TOF-MS/MS was also undertaken and representative examples of fragmentation are given in Supplementary Fig. 10. The major MS/MS fragment of all compounds was m/z 144, which corresponded to the loss of the vancosamine. This and other fragments showed that the two sugars, vancosamine and glucose, remained intact throughout the reaction sequence. Sequential losses of the MIE and EEPS fragments could also be observed in the MS/MS spectra. ¹H, ¹³C and 2D (COSY, TOCSY, HSQC, HMBC and HSQC-TOCSY) NMR experiments were used to confirm the structure of **19** (Supplementary Table 1).

### Synthesis of 2-[(2-Pyridinyl)dithio]ethanamino-vancomycin (43):
To a solution of vancomycin hydrochloride **1** (6.89 g, 4.64 mmol, 1.2 equiv) in dry dimethylsulphoxide (DMSO) (68 mL, 69 mM) was added a solution of (S)-pyridylthio cysteamine hydrochloride (PDEA, 998 mg, 3.87 mmol) in dry DMF (68 mL, 56.9 mM). PyBOP (2.82 g, 5.42 mmol, 1.4 equiv) in dry DMF (10.8 mL, 0.5 M) was added followed by DIPEA (3.37 mL, 19.3 mmol, 5 equiv). The resulting solution was stirred at room temperature overnight and the solvent was removed under reduced pressure. The crude product was precipitated with diethyl ether and collected by vacuum filtration as a white solid, 10.8 g. The crude product was used for the disulphide exchange reaction without further purification. A small amount of crude product (100 mg) was purified by HPLC to give **43** as white solid (TFA salt), 47 mg, 71% yield. HPLC purity > 95%, $t_R$ 6.4. (ES) m/z (MH₂²⁺) 808.7, (MH₃³⁺) 540.0. HRMS exact mass (ESI microTOF-LC): calcd for C₇₃H₈₅Cl₂N₁₁O₂₃S₂ 808.7314 (MH₂²⁺), found 808.7348.

### nC9CO-GSSKSPSKKKKKKPGDC(SCH₂CH₂NH-Vancomycin)-NH₂ (6):
HPLC purity > 95%, $t_R$ 4.8. (ES) m/z (MH₃³⁺) 1150.3, (MH₄⁴⁺) 863.0 (MH₅⁵⁺) 690.7, (MH₆⁶⁺) 575.8, (MH₇⁷⁺) 493.7.

### nC11CO-GSSKSPSKKKKKKPGDC(SCH₂CH₂NH-Vancomycin)-NH₂ (7):
HPLC purity > 95%, $t_R$ 5.0. (ES) m/z (MH₃³⁺) 1159.4, (MH₄⁴⁺) 870.2 (MH₅⁵⁺) 696.3, (MH₆⁶⁺) 580.5, (MH₇⁷⁺) 497.6. HRMS exact mass (ESI microTOF-LC): calcd for C₁₅₅H₂₄₁Cl₂N₃₅O₄₇S₂⁴⁺ 869.6585 (MH₄⁴⁺), found 869.6595.

### Myristoyl-GSSKSPSKKKKKKPGDC-NH₂ (8a):
HPLC purity > 95%, $t_R$ 5.5. (ES) m/z (MH₂²⁺) 1000.0, [M + 3H]³⁺ 667.2, [M + 4 H]⁴⁺ 500.7, [M + 5 H]⁵⁺ 400.8, HRMS exact mass (ESI microTOF-LC): calcd for C₈₉H₁₆₇N₂₅O₂₄S⁴⁺ 500.5579 (MH₄⁴⁺), found 500.5645.

### nC13CO-GSSKSPSKKKKKKPGDC(SCH₂CH₂NH-Vancomycin)-NH₂ (8):
HPLC purity > 95%, $t_R$ 5.0. (ES) m/z (MH₃³⁺) 1168.9, (MH₄⁴⁺) 877.2 (MH₅⁵⁺) 702.0, (MH₆⁶⁺) 585.2, (MH₇⁷⁺) 501.7. HRMS exact mass (ESI microTOF-LC): calcd for C₁₅₇H₂₄₈Cl₂N₃₅O₄₇S₂⁷⁺ 501.3839 (MH₇⁷⁺), found 501.3819.

### nC15CO-GSSKSPSKKKKKKPGDC(SCH₂CH₂NH-Vancomycin)-NH₂ (9):
HPLC purity > 95%, $t_R$ 5.3. (ES) m/z (MH₃³⁺) 1178.2, (MH₄⁴⁺) 884.1 (MH₅⁵⁺) 707.5, (MH₆⁶⁺) 589.8, (MH₇⁷⁺) 505.7. HRMS exact mass (ESI microTOF-LC): calcd for C₁₅₉H₂₅₂Cl₂N₃₅O₄₇S₂⁷⁺ 505.3884 (MH₇⁷⁺), found 505.3859.

### 4-Ph-Bz-GSSKSPSKKKKKKPGDC(SCH₂CH₂NH-Vancomycin)-NH₂ (10):
HPLC purity > 95%, $t_R$ 4.8. (ES) m/z (MH₃³⁺) 1159.2, (MH₄⁴⁺) 869.6, (MH₅⁵⁺) 695.8, (MH₆⁶⁺) 580.1, (MH₇⁷⁺) 497.3.

### nC13CO-GSSKSPSKKKKKKPGDC(SCH₂CH₂NH-Vancomycin)-NH₂ (11):
HPLC purity > 95%, $t_R$ 6.1. (ES) m/z (MH₃³⁺) 1040.8, (MH₄⁴⁺) 781.0 (MH₅⁵⁺) 625.0, (MH₆⁶⁺) 521.0. HRMS exact mass (ESI microTOF-LC): calcd for C₁₃₉H₂₁₁Cl₂N₂₉O₄₄S₂⁶⁺ 520.7325 (MH₆⁶⁺), found 520.7328.

**Myristoyl-GSSKSPSKKKPC-NH$_2$ (12a)**: HPLC purity > 95%, $t_R$ 5.7. (ES) $m/z$ (MH$_2^{2+}$) 721.9, [M + 3 H]$^{3+}$ 481.7, [M + 4 H]$^{4+}$ 361.5. HRMS exact mass (ESI microTOF-LC): calcd for C$_{65}$H$_{121}$N$_{17}$O$_{17}$S$_2^{2+}$ 721.9418 (MH$_2^{2+}$), found 721.9427.

**nC13CO-GSSKSPSKKKPC(SCH$_2$CH$_2$NH-Vancomycin)-NH$_2$ (12)**: HPLC purity > 95%, $t_R$ 5.2. (ES) $m/z$ (MH$_2^{2+}$) 1474.1, (MH$_3^{3+}$) 983.6, (MH$_4^{4+}$) 738.0 (MH$_5^{5+}$) 590.2, (MH$_6^{6+}$) 492.4. HRMS exact mass (ESI microTOF-LC): calcd for C$_{133}$H$_{202}$Cl$_2$N$_{27}$O$_{40}$S$_2^{5+}$ 590.2679 (MH$_5^{5+}$), found 590.2701.

**Myristoyl-GSKKKC-NH$_2$ (13a)**: HPLC purity > 95%, $t_R$ 6.1. (ES) $m/z$ (MH$^+$) 859.4, (MH$_2^{2+}$) 430.3, [M + 3 H]$^{3+}$ 287.3. HRMS exact mass (ESI microTOF-LC): calcd for C$_{40}$H$_{80}$N$_{10}$O$_8$S$^{2+}$ 430.2935 (MH$_2^{2+}$), found 430.3061.

**nC13CO-GSKKKC(SCH$_2$CH$_2$NH-Vancomycin)-NH$_2$ (13)**: HPLC purity > 95%, $t_R$ 5.8. (ES) $m/z$ (MH$_2^{2+}$) 1182.4, (MH$_3^{3+}$) 789.0, (MH$_4^{4+}$) 592.1 (MH$_5^{5+}$) 473.9. HRMS exact mass (ESI microTOF-LC): calcd for C$_{108}$H$_{160}$Cl$_2$N$_{20}$O$_{31}$S$_2^{4+}$ 591.7589 (MH$_4^{4+}$), found 591.7585.

**Myristoyl-GSKKKC-OH (14a)**: HPLC purity > 95%, $t_R$ 6.3. (ES) $m/z$ (MH$^+$) 860.4, (MH$_2^{2+}$) 430.8, (MH$_3^{3+}$) 287.6. HRMS exact mass (ESI microTOF-LC): calcd for C$_{40}$H$_{78}$N$_9$O$_9$S$^+$ 860.5638, found 860.5802.

**nC13CO-GSKKKC(SCH$_2$CH$_2$NH-Vancomycin)-OH (14)**: HPLC purity > 95%, $t_R$ 5.9. (ES) $m/z$ (MH$_2^{2+}$) 1184.7, (MH$_3^{3+}$) 789.7, (MH$_4^{4+}$) 592.4 (MH$_5^{5+}$) 474.1. HRMS exact mass (ESI microTOF-LC): calcd for C$_{108}$H$_{158}$Cl$_2$N$_{19}$O$_{32}$S$_2^{4+}$ 789.0041 (MH$_4^{4+}$), found 789.0006.

**Myristoyl-KKKC-OH – disulphide dimer (15a)**: HPLC purity > 95%, $t_R$ 6.4. (ES) $m/z$ (MH$^+$) 1429.8, (MH$_2^{2+}$) 715.5, (MH$_3^{3+}$) 477.4, (MH$_4^{4+}$) 358.4 (MH$_5^{5+}$) 286.9. HRMS exact mass (ESI microTOF-LC): calcd for C$_{70}$H$_{138}$N$_{14}$O$_{12}$S$_2^{2+}$ 715.5025, found 715.5058.

**nC13CO-KKKC(SCH$_2$CH$_2$NH-Vancomycin)-OH (15)**: HPLC purity > 95%, $t_R$ 5.8. (ES) $m/z$ (MH$_2^{2+}$) 1110.8, (MH$_3^{3+}$) 741.5, (MH$_4^{4+}$) 556.4 (MH$_5^{5+}$) 445.4. HRMS exact mass (ESI microTOF-LC): calcd for C$_{103}$H$_{149}$Cl$_2$N$_{17}$O$_{29}$S$_2^{2+}$ 1110.9757 (MH$_2^{2+}$), found 1110.9783.

**nC13CO-GSKKK-K(Vancomycin)-OH (16)**: HPLC purity > 95%, **$t_R$** 6.1. (ES) $m/z$ (MH$_2^{2+}$) 772.9, (MH$_4^{4+}$) 579.8, (MH$_5^{5+}$) 464.1. HRMS exact mass (ESI microTOF-LC): calcd for C$_{109}$H$_{159}$Cl$_2$N$_{19}$O$_2^{2+}$ 1158.0382 (MH$_2^{2+}$), found 1158.0396.

**nC13CO-KKK-K(Vancomycin)-OH (17)**: HPLC purity > 95%, $t_R$ 5.9. (ES) $m/z$ 1086.9 (MH$_2^{2+}$), (MH$_3^{3+}$) 724.8, (MH$_4^{4+}$) 543.8. HRMS exact mass (ESI microTOF-LC): calcd for C$_{104}$H$_{153}$Cl$_2$N$_{17}$O$_{29}^{4+}$ 543.5094 (MH$_4^{4+}$), found 543.5109.

**nC10CO-KKK-K(Vancomycin)-OH (18)**: HPLC purity > 95%, $t_R$ 5.5. (ES) $m/z$ (MH$_2^{2+}$) 1065.8, (MH$_3^{3+}$) 710.7, (MH$_4^{4+}$) 533.4. HRMS exact mass (ESI microTOF-LC): calcd for C$_{101}$H$_{146}$Cl$_2$N$_{17}$O$_{29}^{3+}$ 710.3278 (MH$_3^{3+}$), found 710.3245.

**nC10CO-KK-K(Vancomycin)-OH (19)**: HPLC purity > 95%, $t_R$ 4.5. (ES) $m/z$ (MH$_2^{2+}$) 1001.9, (MH$_3^{3+}$) 668.4, (MH$_4^{4+}$) 501.3. HRMS exact mass (ESI microTOF-LC): calcd for C$_{95}$H$_{135}$Cl$_2$N$_{15}$O$_{28}^{4+}$ 500.9739 (MH$_4^{4+}$), found 500.9753.

**nC10CO-K(Vancomycin)-OH (20)**: HPLC purity > 95%, $t_R$ 7.1. (ES) $m/z$ (MH$_2^{2+}$) 873.5, (MH$_3^{3+}$) 582.9. HRMS exact mass (ESI microTOF-LC): C$_{83}$H$_{110}$Cl$_2$N$_{11}$O$_{26}^{3+}$ 582.2328 (MH$_3^{3+}$), found 582.2322.

**nC10CO-KK-K(Vancomycin)-NH2 (21)**: HPLC purity > 95%, $t_R$ 5.4. (ES) $m/z$ (MH$_2^{2+}$) 1000.3, (MH$_3^{3+}$) 667.8, (MH$_4^{4+}$) 501.0. HRMS exact mass (ESI microTOF-LC): calcd for C$_{95}$H$_{136}$Cl$_2$N$_{16}$O$_{27}^{4+}$ 500.7279 (MH$_4^{4+}$), found 500.7270.

**nC10CO-KK-K(Vancomycin)-NHMe (22)**: HPLC purity > 95%, $t_R$ 5.6. (ES) $m/z$ (MH$_2^{2+}$) 1007.8, (MH$_3^{3+}$) 672.3, (MH$_4^{4+}$) 504.5. HRMS exact mass (ESI microTOF-LC): calcd for C$_{96}$H$_{138}$Cl$_2$N$_{16}$O$_{27}^{4+}$ 504.2318 (MH$_4^{4+}$), found 504.2315.

**POP2K-KK-K(Vancomycin)-OH (23)**: HPLC purity > 95%, $t_R$ 6.7. (ES) $m/z$ (MH$_2^{2+}$) 1177.3, (MH$_3^{3+}$) 785.3, (MH$_4^{4+}$) 589.4. HRMS exact mass (ESI microTOF-LC): calcd for C$_{116}$H$_{141}$Cl$_2$N$_{17}$O$_{32}^{3+}$ 1176.9647 (MH$_3^{3+}$), found 1176.9644.

**POB2K-KK-K(Vancomycin)-NHMe (24)**: HPLC purity > 95%, $t_R$ 6.1. (ES) $m/z$ (MH$_2^{2+}$) 1185.2, (MH$_3^{3+}$) 789.9, (MH$_4^{4+}$) 592.6. HRMS exact mass (ESI microTOF-LC): calcd for C$_{117}$H$_{144}$Cl$_2$N$_{18}$O$_{31}^{2+}$ 1183.4806 (MH$_2^{2+}$), found 1183.4811.

**Ac-KK-K(Vancomycin)-OH (25)**: HPLC purity > 95%, $t_R$ 4.6. (ES) $m/z$ (MH$_2^+$) 939.7, (MH$_3^{3+}$) 626.0, (MH$_4^{4+}$) 469.8. HRMS exact mass (ESI microTOF-LC): calcd for C$_{86}$H$_{117}$Cl$_2$N$_{15}$O$_{28}^{4+}$ 469.4387 (MH$_4^{4+}$), found 469.4365.

**POB2K-K-Dap-K(Vancomycin)-NHMe (27)**: HPLC purity > 95%, $t_R$ 5.3. (ES) $m/z$ (MH$_2^{2+}$) 1162.4, (MH$_3^{3+}$) 775.8, (MH$_4^{4+}$) 582.1. HRMS exact mass (ESI microTOF-LC): calcd for calcd for C$_{114}$H$_{139}$Cl$_2$N$_{18}$O$_{31}^{3+}$ 775.3071 (MH$_3^{3+}$), found 775.3041.

**POB2K-Dap-K-K(Vancomycin)-NHMe (28)**: HPLC purity > 95%, $t_R$ 5.2. (ES) $m/z$ (MH$_2^{2+}$) 1163.8, (MH$_3^{3+}$) 775.8, (MH$_4^{4+}$) 582.1. HRMS exact mass (ESI microTOF-LC): calcd for C$_{114}$H$_{139}$Cl$_2$N$_{18}$O$_{31}^{3+}$ 775.3071 (MH$_3^{3+}$), found 775.3042.

**POB2K-Dap-Dap-K(Vancomycin)-NHMe (29)**: HPLC purity > 95%, $t_R$ 5.4. (ES) $m/z$ (MH$_2^{2+}$) 1142.7, (MH$_3^{3+}$) 761.9, (MH$_4^{4+}$) 571.6. HRMS exact mass (ESI microTOF-LC): calcd for C$_{111}$H$_{133}$Cl$_2$N$_{18}$O$_{31}^{3+}$ 761.2915 (MH$_3^{3+}$), found 761.2932.

**POB2K-K-Dab-K(Vancomycin)-NHMe (30)**: HPLC purity > 95%, $t_R$ 5.2. (ES) $m/z$ (MH$_2^{2+}$) 1169.4, (MH$_3^{3+}$) 780.5, (MH$_4^{4+}$) 585.7. HRMS exact mass (ESI microTOF-LC): calcd for C$_{115}$H$_{141}$Cl$_2$N$_{18}$O$_{31}^{3+}$ 779.9790 (MH$_3^{3+}$), found 779.9766.

**POB2K-Dab-K-K(Vancomycin)-NHMe (31)**: HPLC purity > 95%, $t_R$ 5.4. (ES) $m/z$ (MH$_2^{2+}$) 1169.9, (MH$_3^{3+}$) 780.5, (MH$_4^{4+}$) 585.6. HRMS exact mass (ESI microTOF-LC): calcd for C$_{115}$H$_{141}$Cl$_2$N$_{18}$O$_{31}^{3+}$ 779.9790 (MH$_3^{3+}$), found 779.9761.

**POB2K-Orn-Orn-K(Vancomycin)-NHMe (34)**: HPLC purity > 95%, $t_R$ 5.1. (ES) $m/z$ (MH$_2^{2+}$) 1170.7, (MH$_3^{3+}$) 779.9, (MH$_4^{4+}$) 585.6. HRMS exact mass (ESI microTOF-LC): calcd for C$_{115}$H$_{141}$Cl$_2$N$_{18}$O$_{31}^{3+}$ 779.9790 (MH$_3^{3+}$), found 779.9788.

**POB2K-K-Orn-K(Vancomycin)-NHMe (32)**: HPLC purity > 95%, $t_R$ 5.1. (ES) $m/z$ (MH$_2^{2+}$) 1176.8, (MH$_3^{3+}$) 785.0, (MH$_4^{4+}$) 589.2. HRMS exact mass (ESI microTOF-LC): calcd for C$_{116}$H$_{144}$Cl$_2$N$_{18}$O$_{31}^{4+}$ 588.7400 (MH$_4^{4+}$), found 588.7418.

**POB2K-Orn-K-K(Vancomycin)-NHMe (33)**: HPLC purity > 95%, $t_R$ 5.2. (ES) $m/z$ (MH$_2^{2+}$) 1177.8, (MH$_3^{3+}$) 785.2, (MH$_4^{4+}$) 589.2. HRMS exact mass (ESI microTOF-LC): calcd for C$_{116}$H$_{143}$Cl$_2$N$_{18}$O$_{31}^{3+}$ 784.6509 (MH$_3^{3+}$), found 784.6490.

**POB2K-KK-Orn(Vancomycin)-NHMe (35)**: HPLC purity > 95%, $t_R$ 6.1. (ES) $m/z$ (MH$_2^{2+}$) 1177.8, (MH$_3^{3+}$) 785.2, (MH$_4^{4+}$) 589.1. HRMS exact mass (ESI microTOF-LC): calcd for C$_{116}$H$_{143}$Cl$_2$N$_{18}$O$_{31}^{3+}$ 784.6509 (MH$_3^{3+}$), found 784.6497.

**nC9CO-KKK-K(Vancomycin)-OH (36)**: HPLC purity > 95%, $t_R$ 5.2. (ES) $m/z$ (MH$_2^{2+}$) 1059.2, (MH$_3^{3+}$) 706.2, (MH$_4^{4+}$) 529.8. HRMS exact mass (ESI microTOF-LC): calcd for C$_{100}$H$_{145}$Cl$_2$N$_{17}$O$_{29}$ 529.4937 (MH$_4^{4+}$), found 529.4933.

*nC11CO-KKK-K(Vancomycin)-OH (37)*. HPLC purity > 95%, $t_R$ 5.6. (ES) $m/z$ (MH$_2^{2+}$) 1071.9, (MH$_3^{3+}$) 715.0, (MH$_4^{4+}$) 536.9. HRMS exact mass (ESI microTOF-LC): calcd for C$_{102}$H$_{147}$Cl$_2$N$_{17}$O$_{29}$ 1071.9958 (MH$_2^{2+}$), found 1072.0001.

**nC12CO-KKK-K-(Vancomycin)-OH (38)**: HPLC purity > 95%, $t_R$ 5.7. (ES) $m/z$ (MH$_2^{2+}$) 1078.9, (MH$_3^{3+}$) 719.9, (MH$_4^{4+}$) 540.3. HRMS exact mass (ESI microTOF-LC): calcd for C$_{103}$H$_{150}$Cl$_2$N$_{17}$O$_{29}$ 719.6715 (MH$_3^{3+}$), found 719.6710.

**nC10CO-GSKKK-K-(Vancomycin)-OH (39)**: HPLC purity > 95%, $t_R$ 5.4. (ES) $m/z$ (MH$_2^{2+}$) 1136.9, (MH$_3^{3+}$) 758.7, (MH$_4^{4+}$) 569.1. HRMS exact mass (ESI microTOF-LC): calcd for C$_{106}$H$_{154}$Cl$_2$N$_{19}$O$_{32}$ 758.3456 (MH$_3^{3+}$), found 758.3477.

**POB-KKK-K-(Vancomycin)-OH (40)**: HPLC purity > 95%, $t_R$ 5.0. (ES) $m/z$ (MH$_2^{2+}$) 1079.3, (MH$_3^{3+}$) 720.2, (MH$_4^{4+}$) 540.4. HRMS exact mass (ESI microTOF-LC): calcd for C$_{103}$H$_{133}$Cl$_2$N$_{17}$O$_{30}^{2+}$ 1078.9385 (MH$_2^{2+}$), found 1078.9326.

**POB-KK-K-(Vancomycin)-OH (41)**: HPLC purity > 95%, $t_R$ 5.9. (ES) $m/z$ (MH$_2^{2+}$) 1015.3, (MH$_3^{3+}$) 677.3, (MH$_4^{4+}$) 508.3. HRMS exact mass (ESI microTOF-LC): calcd for C$_{97}$H$_{123}$Cl$_2$N$_{15}$O$_{29}^{4+}$ 507.9492 (MH$_4^{4+}$), found 507.9498.

**POB2K-KKK-K-(Vancomycin)-OH (42)**: HPLC purity > 95%, $t_R$ 6.0. (ES) $m/z$ (MH$_2^{2+}$) 1241.3, (MH$_3^{3+}$) 828.3, (MH$_4^{4+}$) 621.5. HRMS exact mass (ESI microTOF-LC): calcd for C$_{122}$H$_{155}$Cl$_2$N$_{19}$O$_{33}^{4+}$ 621.0097 (MH$_4^{4+}$), found 621.0068.

**Determination of antimicrobial activity.** Antimicrobial activity of compounds was tested against a number of bacterial strains by broth microdilution (BMD) assay with MIC determination, including *S. aureus* (MSSA ATCC® 29213™, MRSA ATCC® 43300™, VISA NARSA NRS1, VISA NARSA NRS4), *Streptococcus pneumoniae* (MDR ATCC® 700677™) and *Enterococcus faecium* (MDR VanA ATCC® 51559™). ATCC strains were sourced from the American Type Culture Collection and NRS strains from NARSA (Network on Antimicrobial Resistance in *S. aureus*) via BEI Resources (www.beiresources.org). All compounds were prepared to 160 μg mL$^{-1}$ solution in water from a stock solution of 1 mM concentration.

**MIC assay.** The compounds, along with standard antibiotics were serially diluted twofold across the wells of 96-well non-binding surface (NBS) polystyrene plates (Corning 3641). (Note: We have found that MIC determinations in NBS plates give identical results to untreated polystyrene plates for most antibiotics (e.g. vancomycin, daptomycin, β-lactams, oxazolidinones), but for lipoglycopeptide antibiotics (dalbavancin, oritavancin and our vancapticins) where Clinical & Laboratory Standards Institute (CLSI) guidelines recommend addition of 0.002% polysorbate 80 for MIC determination, values obtained in NBS plates with no additive match those obtained in untreated polystyrene plates with addition of 0.002% polysorbate 80—Supplementary Table 12). Standards ranged from 64 to 0.03 μg mL$^{-1}$ and compounds from 8 to 0.003 μg mL$^{-1}$ with final assay volumes of 100 μL per well. Gram-positive bacteria were cultured in Mueller Hinton broth (MHB) (Bactolaboratories, Cat. no. 211443) at 37 °C overnight. A sample of each culture was then diluted 40-fold in fresh MHB and incubated at 37 °C for 2–3 h. The resultant mid-log phase cultures were added to the compound-containing 96-well plates to give a final cell density of $5 \times 10^5$ colony forming unit (CFU) mL$^{-1}$. All the plates were covered and incubated at 37 °C for 24 h. MICs were the lowest concentration showing no visible growth.

**Serum reversal MIC assay.** The compounds, along with standard antibiotics, tested in BMD-MIC assay as above, in the presence of a mixture of 50% human serum (Sigma-Aldrich) and 50% MHB.

**Surfactant reversal MIC assay.** The compounds, along with standard antibiotics, were tested in BMD-MIC assay as above, in the presence of 1 and 5% Survanta (beractant) suspension (Abbott Australasia Cat # 1039.008; 25 mg mL$^{-1}$) in MHB.

**Lipid II ligand antagonism assay.** All compounds were dissolved in $H_2O$. *Bacillus subtilis* ATCC® 6633™ NRS-231 was cultured in MHB at 37 °C for 48 h. A sample of each culture was then diluted 40-fold in fresh MHB and incubated at 37 °C for 18 h. Compounds were added to each row of 96-well NBS plates (NBS, Corning 3641), in duplicate. The tripeptide *N,N′*-diacetyl-L-Lys-D-Ala-D-Ala was serially diluted twofold across the compound-containing wells from 6400 to 3 µg mL$^{-1}$. To this the resultant mid-log phase cultures were diluted to the final concentration of $5 \times 10^5$ CFU mL$^{-1}$, then added to wells of the compound-containing plates. All the plates were covered and incubated at 37 °C for 24 h. The molar excess was determined as the ratio of [ligand]/[antibiotic] in the well containing the lowest ligand concentration that completely suppressed inhibition after 24 h incubation.

**Time kill assay.** *S. aureus* ATCC® 43300™ (MRSA) was cultured in MHB (BactoLaboratories, Cat. No 211443) at 37 °C overnight. A sample was then diluted 40-fold in fresh MHB and incubated at 37 °C for 2–3 h until the culture reached early exponential phase (OD$_{600}$ of 0.2–0.4). Vancapticin **24** (1, 0.25, 0.015 and 0.003 µg mL$^{-1}$) and vancomycin (4, 2, 1 and 0.5 µg mL$^{-1}$) were serially diluted twofold across the wells of 96-well microtitre plates (NBS, Corning 3641) with a final volume of 50 µL, then combined with 50 µL of bacterial inoculum (giving a final cell density of $5 \times 10^5$ CFU mL$^{-1}$) and incubated at 37 °C. Serial dilutions of the cultures were plated at each desired time point, using a multichannel pipette as follows: in row A of a 96-well plate, 50 µL of sterile activated charcoal suspension (25 mg mL$^{-1}$) was added, while 90 µL of 0.9% sterile saline per well was added to the other rows. At selected time-points, 50 µL aliquots were transferred from the time-kill assay plate to the first row containing the charcoal suspension and mixed well. The wells were further diluted 1 in 10 (0.9% saline) for the appropriate number of dilutions and 10 µL of each dilution was spotted in duplicate onto Luria-Bertani (LB) agar, then incubated overnight at 37 °C. The colonies in each spot were counted and used to calculate the number of viable CFU mL$^{-1}$ remaining in the original culture by considering the dilution factors (1:2 in charcoal, the serial dilution factor and the volume of the aliquot spotted).

**Resistance frequency assay.** The resistance frequency for Compound **19** was determined following literature procedures[41, 42].

**Compound preparation:** Compound **19** was prepared to 160 µg mL$^{-1}$ solution in water from the given stock solution of 1 mM concentration. Vancomycin was prepared to 1.28 mg mL$^{-1}$ solution in water.

**Antibacterial plate preparation:** Tryptic soy broth (TSB) with 1% agarose and 0.002% Tween 80 was maintained at 45 to 50 °C in a water bath. 160 mL of agarose was transferred to 200 mL glass bottle, maintaining the remaining agarose in the water bath. The appropriate amount of compound (calculated to give a final concentration 16-fold or 8-fold over MIC) was added to the 160 mL agarose sample [A]. A volume of 80 mL of agarose sample [A] was plated in four petri plates (duplicates/condition—neat, 10$^{-1}$) with swirling to ensure that the bottom of the plate was coated equally. An additional 80 mL of fresh agarose was added to the remaining 80 mL of agarose [A] sample to achieve a twofold dilution to provide agarose sample [B]. Once again, 80 mL of agarose sample [B] was plated in four petri plates. This procedure was repeated until a range of concentrations of the antibacterial compound above the expected MIC was obtained. This method was done individually for every antibacterial compound under study. Agarose plate MIC values used for this study were determined to be 1 µg mL$^{-1}$ for vancomycin and 0.06 µg mL$^{-1}$ for compound **19**.

**Plate culture for titre of 18 h stationary phase and resistance frequency:** Bacteria were subcultured at 37 °C overnight with shaking at 220 rpm. The bacterial culture was serially diluted in MHB to achieve a 10$^{-1}$ to 10$^{-8}$ dilution series. The three last dilutions (10$^{-6}$, 10$^{-7}$ and 10$^{-8}$) were used to determine the titre of the culture. An aliquot of 100 µL (10 µL in some cases) of these three dilutions were added to compound-free agar petri plates. This was done in duplicate and the plates were incubated overnight at 37 °C. Resultant separated visible colonies were counted; those colonies that grow together were not included to determine the titre of the culture. An aliquot of 100 µL of the neat (undiluted) stationary-phase culture and 10$^{-1}$ dilution was plated separately onto the agar plate containing each concentration of antibacterial compound. The plates were incubated overnight at 37 °C.

**Detection and analysis:** The colonies were counted after 24 h. The resistance frequency is determined by dividing the number of colonies obtained on the plates containing the antibacterial agent by the number of cells plated.

**Resistance induction assay.** A resistance induction assay was conducted for vancomycin, daptomycin, compound **17** and compound **24**[43].

*S. aureus* ATCC® 43300™ was grown in Mueller Hinton broth supplemented with 50 µg mL$^{-1}$ of CaCl$_2$. The bacterial culture was incubated at 37 °C in a shaker at 195 rpm.

**Viability testing:** The mid log *S. aureus* growth culture was serially diluted and plated on a solid LB agar plates in quadruplicates and incubated at 37 °C overnight to determine viable colony count.

**Antimicrobial broth microdilution MICs and plate preparation:** The initial determination of the compound MICs were taken from the literature and empirically measured. Doubling MIC dilutions were prepared in Cation-adjusted Mueller-Hinton broth (Ca-MHB) in NBS 96-well plates (NBS, Corning 3641) with columns 11 and 12 allocated for bacterial culture without the antimicrobial and Ca-MHB media. The starting MIC ranges were 20 µg mL$^{-1}$ to 0.08 µg mL$^{-1}$ for daptomycin, 32 µg mL$^{-1}$ to 0.06 µg mL$^{-1}$ for vancomycin and 0.5 µg mL$^{-1}$ to 0.001 µg mL$^{-1}$ for both vancapticins with 100 µL of the dilutions added to the plate. MRSA culture in mid log growth with 0.6 optical density at 600 nm (OD$_{600}$) was diluted to an equivalence of 10$^5$ CFU mL$^{-1}$ and 100 µL was added to the antimicrobial containing plate. The plate was then incubated at 37 °C overnight.

**Daily antimicrobial MIC plate preparation:** After overnight incubation, the growth of the *S. aureus* (MRSA) culture was determined by reading at OD$_{600}$ on the Epoch microplate spectrophotometer and 0.1 OD$_{600}$ was the chosen cut-off determining growth vs. no growth. The sample selection of *S. aureus* and preparation was carried out as previously described[43] with 10 µL of the sub-lethal MIC *S. aureus* cultures (i.e. the maximum antibiotic concentration with OD$_{600} \geq 0.1$) diluted in 10 mL of Ca-MHB for the following day's MIC plate. Doubling dilutions were supplemented with intermediate MICs to reduce the MIC difference and provide a ladder for non-susceptible isolates for the 20 day period with an average overnight incubation of 19 h at 37 °C. The non-susceptible isolates were grown overnight at 37 °C with shaking before being preserved in 20% glycerol and as a pellet (stored frozen at −80 °C).

**Cytotoxicity assay.** Compound cytotoxicity was evaluated against HepG2 (ATCC®-HB-8065 liver hepatoceullular carcinoma) and HEK293 (ATCC®CRL-1573, embryonic kidney) human cell lines using standard (3-(4,5-dimethylthiazol-2-yl)−2,5-diphenyltetrazolium bromide) (MTT) assay protocol with slight changes. In brief, HepG2 and HEK293 cells (assessed annually for mycoplasma contamination) were seeded as $1.5 \times 10^4$ cells per well in a clear 96-well tissue culture treated plate in a final volume of 100 µL in Dulbecco's Modified Eagle Medium (DMEM) (GIBCO-Invitrogen #11995-073), in which 1% of FBS was added. Cells were incubated for 24 h at 37 °C, 5% CO$_2$ to allow cells to attach to the plates. Compounds prepared at 100 µM to 0.046 µM in threefold dilutions were added into each well and incubated for 24 h at 37 °C, 5% CO$_2$. After the incubation, 0.4 mg mL$^{-1}$ MTT (3-(4,5-dimethylthiazol-2-yl)−2,5-diphenyltetrazolium bromide) (Invitrogen) was added to each well. The plates were then incubated for 2 h at 37 °C, 5% CO$_2$. The media was removed, and purple formazan crystals were resuspended in 60 µL of DMSO. The absorbance was read at 570 nm using a Polarstar Omega plate reader. The data was then analysed by Prism software. Results are presented as the average percentage of control ± S.D. for each set of duplicate wells using the following equation: percent viability = (ABS$_{TEST}$ − ABS$_{POSITIVE \ CTRL}$ / ABS$_{UNTREATED}$ −ABS$_{POSITIVE \ CTRL}$)×100.

**Haemolysis assay.** Haemolytic activity was measured as previously described with slight modifications[44, 45]. Fresh human venal blood was collected from healthy volunteers and pretreated with the anticoagulant sodium citrate. Serial twofold dilutions of each antibiotic, dissolved in 0.9% NaCl solution, were added to fresh human whole blood (50% final, vol:vol) to the final volume of 80 µL and incubated at 37 °C for 3 h on a horizontal shaker at 200 rpm. For negative and positive controls, human whole blood in 0.9% NaCl solution ($A_{blank}$) and in sdH$_2$O ($A_{water}$) were used, respectively. After incubation, both test and $A_{blank}$ were mixed with 1 mL 0.9% NaCl solution, or sdH$_2$O for the complete hydrolysis $A_{water}$. After centrifugation of samples at 2500 rpm (700 × $g$) for 5 min, the release of haemoglobin was monitored by measuring the absorbance ($A_{sample}$) of the supernatant at 540 nm using BMG Labtech PolarStar Omega multimode reader. The percentage of haemolysis was calculated according to the equation:

$$\% \text{ haemolysis} = \left[ \left( A_{sample} - A_{blank} \right) / \left( A_{water} - A_{blank} \right) \right] \times 100$$

**Critical micelle concentration assay.** CMCs were determined as previously described[46] with a higher throughput modification. Briefly, a cascade dilution of each antibiotic was performed in a low binding polypropylene V-bottom shaped 96-well plates in the presence of 0.2 M NaCl solution across 12 concentrations ranging from 1000 to 0.1 µM, except the control melittin (100 to 0.01 µM). An identical cascade dilution was performed for the neutral detergent Triton X-100 and the anionic detergent sodium dodecyl sulphate (SDS) starting from 0.1% down to 0.00001% (vol:vol for Triton X-100 and wt:vol for SDS). An aliquot of 6-diphenyl-1,3,5-hexatriene (DPH) in DMSO was added in each well to yield the final DPH concentration of 5 µM. The reaction mixtures were mixed for 30 s and transferred in horizontal triplicate (25 µL per well) into a black falcon 384-well plate. The 384-well plate was then sealed, wrapped up in aluminium foil and incubated for 30 min at room temperature. The fluorescence intensity was

monitored in real time using a BMG Labtech PolarStar Omega multimode reader fitted with 360-10 and 430-10 excitation and emission filters respectively, at an excitation wavelength of 360 nm and an emission wavelength of 430 nm.

**Membrane depolarisation assay.** Antibiotic-induced bacterial cytoplasmic membrane depolarisation was determined using the fluorescent dye 3,3-dipropylthiacarbocyanine diSC3(5) (Sigma Aldrich, Australia) as previously described[37]. Briefly, early exponential phase cells were pelleted and washed twice in 10 mM HEPES buffer (pH 7.4) containing 50 µg mL$^{-1}$ of CaCl$_2$ and 5 mM Glucose, and then re-suspended in the same buffer ($2 \times 10^8$ CFU mL$^{-1}$). diSC3(5) was added to a final concentration of 1.5 µM and incubated at room temperature in the dark to enable dye uptake and fluorescence quenching. After 30 min, cells were diluted 50-fold in assay buffer and 45 µL of cells were added to a 384-well black walled polystyrene plate (Corning, CLS3573) (Note: NBS plates were not employed, and no polysorbate-80 was added to avoid potential effects on membrane permeability, or interference with fluorescence as previously reported[37]. The MICs of compounds in polystyrene plates are listed in Supplementary Table 12). Background data was collected before adding compounds using a Tecan Infinite® m1000 Pro Multi-mode reader (excitation/emission 612/665 nm) to ensure fluorescence quenching. Compounds were added to a final concentration of 16 µg mL$^{-1}$ and fluorescence was monitored for 30 min. Data were corrected by subtraction of background response (diSC3(5) in the presence of untreated cells, no compound). Note that the MIC values for oritavancin and compound **17** increase to 1 µg mL$^{-1}$, and compound **24** to 2 µg mL$^{-1}$, in polystyrene plates. Each sample was tested in quadruplicate and independent assays were performed twice showing similar results. Data were analysed and presented using Prism software.

**Membrane permeability assay.** To evaluate the ability of the compounds to disrupt the cytoplasmic membrane integrity, the membrane impermeable fluorescent DNA intercalating dye propidium iodide (PI) was used. Early exponential phase cells were pelleted and washed twice in 10 mM HEPES buffer (pH 7.4) containing 50 µg mL$^{-1}$ of CaCl$_2$ and 5 mM glucose, and then re-suspended in the same buffer ($4 \times 10^8$ CFU mL$^{-1}$). Cell suspension was added to a 384-well black walled polystyrene plate (Corning, CLS3573) containing compounds, giving a final concentration of 16 µg mL$^{-1}$. After 1 h incubation at 37 °C, 5 µg mL$^{-1}$ of PI was added and fluorescence was monitored for 90 min using a Tecan Infinite® m1000 Pro Multi-mode reader (excitation/emission 535/620 nm). Data were corrected by subtraction of fluorescence signal arising from untreated cells in the presence of PI. Note that the MIC values for oritavancin and compound **17** increase to 1 µg mL$^{-1}$, and compound **24** to 2 µg mL$^{-1}$, in polystyrene plates. Each sample was tested in quadruplicate and independent assays were performed twice showing similar results. Data was analysed and presented using Prism software.

**ITC for assessment of ligand binding and dimerisation.** Calorimetric experiments were performed on a MicroCal Omega Auto-iTC200 Isothermal Titration Calorimeter (GE Healthcare, Australia), at 298 K, with a 10 µcal s$^{-1}$ reference power and stirring speed of 1000 rpm. All samples were prepared in 0.1 M NaOAc, pH 5.0 buffer. Each experiment consisted of sequential injections into the calorimeter cell (0.2 mL) with 240 s equilibration intervals. Corrections of small heat effects from separate blank titrations were made prior to analysis. Data were processed using the MicroCal Origin 7.0 software package provided with the instrument to fit a single site binding model or dissociation model to estimate the association constant ($K_a$), enthalpy ($\Delta H$), entropy ($\Delta S$) and stoichiometry ($N$). The Gibbs free energy change ($\Delta G$) was calculated using the Gibbs–Helmholz thermodynamic equation, (1):

$$\Delta G = -RT \ln K \qquad (1)$$

Where $R$ is the ideal gas constant (8.31 J mol$^{-1}$ K$^{-1}$) and $T$ is the temperature in Kelvin.

**Single site binding:** Ligand binding experiments consisted of 12 sequential injections (2 µL per injection, first injection of 0.5 µL) of ligand (400 µM, concentration 14-fold to 20-fold higher than the antibiotic concentration) into the calorimeter cell containing the antibiotic (monomer solution 25 µM for vancomycin and 40 µM for vancapticins). Antibiotic concentration was selected to avoid self-dimerisation.

**Dimer-monomer dissociation:** Dissociation experiments involved 13 sequential injections (3 µL per injection, first injection of 0.5 µL) of concentrated (3.0 mM) antibiotic solutions titrated to buffer only. Dissociation experiments involving ligand maintained a constant concentration of excess ligand (9.0 mM) dissolved in buffer in both the calorimeter cell and titration syringe to ensure that antibiotic molecules were predominantly in their ligand-complexed form.

**SPR membrane binding assay.** SPR experiments were performed using a BIAcore 3000 (GE Health) instrument with a Biacore vesicle capture L1 sensor chip. DMPC (1,2-dimyristoyl-*sn*-glycero-3-phosphocholine) and DMPG (1,2-dimyristoyl-*sn*-glycero-3-phosphoglycerol) were purchased from Avanti Polar Lipids. Small unilamellar vesicles (SUV) were prepared in phosphate buffered saline (PBS) (pH 7.4) by sonication and extrusion. In brief, lipids were dissolved in chloroform in 25 mL round-bottom flasks, and then deposited as a thin film by removal of the solvent (chloroform) under reduced pressure on a rotary evaporator and dried under high vacuum for at least 2 h. PBS was then added into each flask to give a suspension, which was sonicated for 5 cycles of 5 min each. The suspension was passed 17 × through a 50-nm polycarbonate filter in an Avestin Lipofast Basic extrusion apparatus to give a translucent solution of vesicles, which should possess a mean diameter of 50 nm. The SUVs were injected across the L1 sensor chip for 30 min at a flow rate of 2 mL min$^{-1}$ to form a supported lipid bilayer. The coverage of the lipid bilayer was determined from the extent of non-specific binding of bovine serum albumin (0.1 mg mL$^{-1}$ in PBS, 5 min injection). A series of concentrations of vancomycin and vancapticins were passed sequentially over the different phospholipid-containing flow cells at a flow rate of 30 µL min$^{-1}$ for 180 s. The amount of antibiotic bound at equilibrium just before the end of the injection was corrected by subtraction of the bulk-refractive index difference observed at the beginning and the end of each injection. After each injection cycle, the lipid surface was regenerated with 20 mM 3-[(3-cholamidopropyl)dimethylammonio]−1-propanesulfonate (CHAPS) and a fresh lipid bilayer was loaded as described above. All assays were carried out at 25 °C in triplicate. The equilibrium binding response values obtained were normalised by dividing the average observed response (RU) by the molecular weights (MW) of each compound, i.e. RU$_{adjusted}$ = 100 × RU/MW.

**Peptidoglycan inhibition assay.** The cell-free particulate fraction of *Bacillus megaterium* KM (ATCC® 13632™) was used to catalyse the polymerisation of peptidoglycan from UPD-linked precursors in vitro following literature procedures[47]. *B. megaterium* was grown in the medium containing 1% (w/v) tryptone (Difco-Bacto), 0.5% (w/v) yeast extract (Difco-Bacto), 0.25% (w/v) K$_2$HPO$_4$ and 0.5% (w/v) glucose with pH 7.2. Bacteria were harvested when the density had reached to 0.4 mg dry weight mL$^{-1}$ and washed in the buffer 0.05 M Tris-HCl (pH 7.8) containing 10 mM MgCl$_2$ by centrifugation at 6000 × g for 10 min at 4 °C. All subsequent manipulations were carried out on ice. The bacteria were resuspended in the same buffer at a density of 50 mg dry weight mL$^{-1}$, and then subjected to three freeze/thaw cycles. Bacteria were then homogenised by using a glass homogeniser at 4 °C (2 × 10 min). The broken cell suspension was centrifuged at 6000 × g for 10 min at 4 °C. The supernatant (Supernatant 1) which contained the unbroken cells and the majority of the cell walls was collected. The pellet was resuspended in the same buffer and re-centrifuged at 6000 × g for 10 min at 4 °C, and the supernatant (Supernatant 2) was collected. The two supernatants (Supernatant 1 and Supernatant 2) were combined and centrifuged at 20,500 rpm (38,000 × g) for 1 h at 4 °C. The pellet, consisting mainly of membrane fragments together with some cell wall material, was washed once with the same buffer and finally resuspended in 0.05 M Tris-HCl (pH 7.8) containing 10 mM MgCl$_2$. In vitro peptidoglycan synthesis was performed by mixing the following contents in a final volume of 20 µL: 50 mM Tris-HCl (pH 7.8), 10 mM MgCl$_2$, *B. megaterium* membrane faction (around 40 to 50 µg protein), 0.4 mM uridine 5'-diphospho *N*-acetyl muramoyl-L-alanyl-D-γ-glutamyl-L-lysyl-D-alanyl-D-alanine (UDP-MurNAc-pentapeptide, The University of Warwick, UK), 6.5 µM [$^{14}$C]UDP-*N*-acetylglucosamine (UDP-[$^{14}$C]GlcNAc, American Radiolabeled Chemicals Inc, 25 µci, 11.1 GBq mmol$^{-1}$, 0.1 mci mL$^{-1}$) and vancapticins at final concentrations of 0, 0.03, 0.1, 1, 3 and 10 µg mL$^{-1}$. The reaction mixtures were incubated at RT for 3 h and placed in a boiling water bath for 3 min to inactivate the enzymes. 5 µL of each sample was separated using TLC on silica gel plates (TLC Silica gel 60 F$_{254}$, 20 × 20 cm, Merck KGaA, Germany) for 2 h in isobutyric acid /1 M NH$_4$OH (5/3, v/v). The TLC plates were then dried and exposed to a phosphorimaging screen (GE Healthcare, Australia) for 1 week and scanned using a Typhoon 8600 (GE Healthcare, Australia). Integrated density value (IDV) of each band on silica gel was analysed using AlphaEase FC software (AlphaImager 2200). The change of peptidoglycan or Lipid II were calculated as percent from the IDV values of the control (without any antibiotic or compound).

**Human plasma stability assay.** Compound stability in human plasma was performed as previously reported[48], with modifications. Briefly, test compounds (20 µM) were prepared from a 0.2 mM H$_2$O stock solution (40 µL in 400 µL assay volume) except eucatropine, which was 300 µg mL$^{-1}$ in 100% DMSO (10 µL in 400 µL assay volume equivalent to 23 µM). The precipitating solution was prepared by adding 5 µL of 10 mM carbutamide in DMSO to 100 mL acetonitrile (final concentration: 135.5 ng mL$^{-1}$ or 0.5 µM carbutamide). The human plasma sample (pooled normal human plasma sodium heparin anticoagulant, 2 µL, Cat No.: HMPLNAHP, BioReclamation) and phosphate-buffer saline (PBS, pH 7.4 isotonic) were pre-heated at 37 °C for 30 min. 40 µL of test compounds (0.2 mM) were added to protein-low binding Eppendorf tubes, 160 µL of buffer was added, and the mixture was vortexed. For eucatropine, 10 µL of eucatropine (300 µg mL$^{-1}$) was added to a protein-low binding Eppendorf tube, 190 µL of buffer was added, and the mixture was vortexed. In each case, 200 µL of plasma was added and the resultant mixture vortexed. For the initial time point (reaction time $t = 0$ h), 50 µL of sample was immediately transferred into a protein-low binding Eppendorf tube and quenched with 150 µL of the ice-cold precipitating solution, followed by

vortexing and subsequent storage at 4 °C. For the remaining time points, the plasma solution was incubated and shaken (150 rpm) at 37 °C. 50 μL aliquots were collected at time points of 0, 1, 3, 6 and 24 h and quenched with 150 μL the ice-cold precipitating solution. All samples were placed in a 4 °C fridge for 30 min then centrifuged at $14,000 \times g$ for 8 min at 4 °C. The supernatant (100 μL) was transferred to a glass vial insert and stored at 4 °C until LC-MS/MS analysis. The percentage of test compounds remaining at the individual time points relative to sample at time point 0 h were reported.

**Glutathione stability assay.** The stability of compounds in the presence of physiological concentrations of glutathione was assessed according to the protocol below: 20 μL of a 1 mM stock solution of the test compound was added to 180 μL of glutathione (reduced form) PBS solution within a plastic HPLC insert, providing a solution with a final concentration of 100 μM of compound and either 5 mM or 0.5 mM in glutathione. The sample was placed in a HPLC sampling rack and sampled at hourly intervals up to 10 h. Care was taken to prepare the sample immediately before injection. The UV area was then plotted for the loss of compound against time. Percentages were plotted relative to total vancomycin derivative at 0 h. Results are shown in Supplementary Fig. 4.

**Microsomal stability assay.** Performed by CDCO, Monash University, Australia.

**Incubation methods:** The solubility of test compounds and their recovery from the incubation matrix were confirmed prior to the metabolic assay. The metabolic stability assay was performed by incubating each test compound (1 μM) with human (Xenotech, Lot #1210057) and mouse (Xenotech, Lot #1110071) liver microsomes at 37 °C and 0.4 mg mL$^{-1}$ protein concentration. The metabolic reaction was initiated by the addition of a nicotinamide adenine dinucleotide phosphate (NADPH)-regenerating system (i.e. NADPH is the cofactor required for cytochrome P450-mediated metabolism) and quenched at various time points over the 60 min incubation period by the addition of acetonitrile. Control samples (containing no NADPH) were included (and quenched at 2, 30 and 60 min) to monitor for potential degradation in the absence of cofactors. Concentrations of each test compound in quenched samples were determined by ultra performance liquid chromatography mass spectrometry (UPLC-MS) (Waters/Micromass Xevo triple quadrupole mass spectrometer). Note: Due to the minimal degradation of test compounds in this assay, a metabolite search was not conducted. In addition, glucuronidation was not considered in this study and may require further assessment using alternative test systems.

**Calculations:** Test compound concentration vs. time data were fitted to an exponential decay function to determine the first-order rate constant for substrate depletion. In cases where clear deviation from first-order kinetics was evident, only the initial linear portion of the profile was utilised to determine the degradation rate constant ($k$). Each substrate depletion rate constant was then used to calculate: (1) a degradation half-life, where $t_{1/2} = \ln(2)\ k^{-1}$; (2) an in vitro intrinsic clearance value, where $CL_{int,in\ vitro} = k$ [microsomal protein content (0.4 mg mL$^{-1}$)]$^{-1}$; (3) a predicted in vivo hepatic intrinsic clearance value, where $CL_{int} = CL_{int,in\ vitro} \times$ [liver mass (g)/body weight (kg)] × [microsomal protein (mg) / liver mass (g)]; and (4) a predicted in vivo hepatic extraction ratio, where $E_H = CL_{blood}\ Q^{-1} = CL_{int}\ [Q + CL_{int}]^{-1}$.

Scaling parameters assumed were: human liver mass 20 g liver / kg body weight[49], human microsomal protein 40 mg g$^{-1}$ liver mass[49, 50], human hepatic blood flow ($Q$) 20.7 mL min$^{-1}$ kg body weight$^{-1}$[51], mouse liver mass 87.5 g liver / kg body weight[51], mouse microsomal protein 45 mg g$^{-1}$ liver mass[51], mouse hepatic blood flow ($Q$) 90 mL min$^{-1}$ kg body weight$^{-1}$ [51].

**In vivo pharmacokinetics.** Performed by WuXi AppTec (Shanghai) Co., Ltd. Seven to 9-week-old male CD1 mice (SLAC Laboratory Animal Co. Ltd., Shanghai, China) weighing 25–35 g were acclimated for approximate 3 days before being used in the study. Animals are group housed during acclimation and in-life study in compliance with the National Research Council 'Guide for the Care and Use of Laboratory Animals.' The animal room environment is controlled (target conditions: temperature 18 to 26 °C, relative humidity 30–70%, 12 h artificial light and 12 h dark) with temperature and relative humidity monitored daily. Animals were deprived of food for approximately 16 h before formulation administration then allowed access to Certified Rodent Diet (Catalogue # M-01F, Shanghai SLAC Laboratory Animal Co. Ltd.) ad libitum 4 h post dosing. Water was autoclaved before being provided to the animals ad libitum. Formulations of compounds were prepared on the morning of the dosing day. The formulation for the IV group was filtered with filter of 0.22 μm before being dosed to animals. After dose formulation preparation, duplicate 50 μL aliquots were removed from each dose formulation for use in dose validation. For each compound studied, three mice were dosed intravenously (IV) administered to each animal via tail vein per facility SOPs using test article formulated in deionised water at 1 mg mL$^{-1}$ with a dose volume of 2 mg mL$^{-1}$, providing a dose of 2 mg mL$^{-1}$ (based on free base concentration). An additional three mice were dosed subcutaneously (SC) administered to each animal via subcutaneous bolus on each animals' back per facility SOPs using test article formulated in deionised water at 2 mg mL$^{-1}$ with a dose volume of 5 mg mL$^{-1}$,

providing a dose of 10 mg mL$^{-1}$. All animals were killed at the last study time point (100% CO$_2$ was introduced into the animal box).

**Sample collection:** Plasma samples were collected as the following target times after each dose administration: IV Sampling Time points (hours post dosing): 0, 0.083, 0.25, 0.5, 1, 2, 4, 8 and 24. SC Sampling Time points (hours post dosing): 0, 0.25, 0.5, 1, 2, 4, 8 and 24. Approximately 30 μL blood was obtained via submandibular or saphenous vein for the first several time points. For the last time point, samples were collected via cardiac puncture while the mouse was under anaesthesia. All blood samples were transferred into pre-chilled plastic microcentrifuge tubes containing 2 μL of ethylenediaminetetraacetic acid dipotassium salt (K2-EDTA) (0.5 M) as anticoagulant and placed on wet ice until centrifugation. Harvested blood samples were centrifuged within 30 min of collection at 7000 rpm ($8546 \times g$) 4 °C for about 10 min. After centrifugation, plasma was transferred into another pre-labelled and pre-chilled polypropylene microcentrifuge tubes, then quick-frozen over dry ice and stored at $-70 \pm 10$ °C until LC/MSMS analysis.

**Sample analysis: Dosing formulations verification:** Aliquots of the formulations were collected in the middle position of each dose formulation in duplicate. A LC-UV method was developed with a calibration curve consisting of six calibration standards. The concentrations of the test compound in dose formulation samples were determined by the LC-UV method. Acceptance criteria for an analytical run: at least five of six calibration standards should be within 20% of nominal values.

**Plasma samples:** LC-MS/MS methods for the quantitative determination of the test article in mouse plasma were developed with an internal standard. Benchtop stability of the compound in mouse plasma was determined at mid QC concentrations in triplicate at 0, 2 h at room temperature. The stability was determined using mean peak area ratio of T2/T0 sample. If the mean peak area ratio is within 80%~120%, the test article in the plasma is considered stable for 2 h at room temperature. A standard curve consists of eight non-zero calibration standards for the LC-MS/MS method with a target lower limit of quantification (LLOQ) at ≤ 3 ng mL$^{-1}$. A set of QC samples consists of three concentration levels (low, middle and high). The sample analysis was performed concurrently with a set of calibration standards and two sets of QC samples using the LC-MS/MS method.

**Acceptance criteria for plasma bioanalytical run:** A minimum of six calibration standards is back calculated to within ± 20% of their nominal concentrations; and a minimum of 4 out of 6 QC samples is back calculated to within ± 20% of their nominal concentrations.

**Analyte interference:** The mean calculated concentration in the single blank matrix should be ≤ 0.5 times the LLOQ.

**Carryover:.** the mean calculated carry-over concentration in the single blank matrix immediately after the highest standard injection should be ≤ LLOQ.

**Data analysis:** Plasma concentration vs. time data from individual animals was analysed by WinNonLin non-compartmental model (Phoenix WinNonlin 6.2.1, Pharsight, Mountain View, CA). Pharmacokinetic parameter $C_0$, $T_{1/2}$, CL, Vd$_{ss}$, $C_{max}$, $T_{max}$, AUC$_{0-last}$, AUC$_{0-inf}$, %F, MRT$_{0-last}$, MRT$_{0-inf}$ and graphs of plasma concentration vs. time profile were derived.

**MRSA thigh infection in vivo efficacy model.** Performed at the University of Queensland, following similar previous literature procedures[52].

Summary: Adult (8-week-old) female CD1 mice were made neutropenic by two injections of cyclophosphamide 4 days and 1 day prior to infection. An inoculum of $10^5$ CFU MRSA (Strain ATCC® 43300™) was injected intramuscularly into both left and right thighs of all mice. Two hours after initiation of infection, saline, vancomycin or vancapticin compounds were injected subcutaneously in the lower back region. After an additional 2 h, a 50 μL sample of blood was obtained from the tail bleed to analyse for presence of antibiotic compound. At 24 h following infection, mice were killed and an additional blood sample collected for compound analysis. Thighs were removed, weighed and homogenised in a fixed volume of saline. The homogenate solution was filtered, diluted and seeded onto agar plates, which wre incubated overnight at 37 °C. Colony counts were used to establish the CFU in the thigh homogenates, the CFU thigh$^{-1}$ and the CFU g$^{-1}$ of thigh.

**Protocol: Compound preparation:** Cyclophosphamide monohydrate (Sigma) was dissolved in sterile saline to a concentration of 30 mg mL$^{-1}$. Likewise, vancomycin (Sigma) and vancapticins were also dissolved in sterile saline to a final concentration of 60 mg mL$^{-1}$ and 18.5 mg mL$^{-1}$ respectively. All compounds were prepared in low binding Eppendorf tubes and kept at $-20$ °C until used.

**Preparation of injectable MRSA solution:** An MRSA subculture bacterial isolate (ATCC$^\circ$43300™) was taken from the storage at −80 °C and freshly seeded on agar plates for overnight growth. From the overnight culture preparation, a single colony was diluted into 10–12 mL of MHB and incubated overnight at 37 °C. A log-phase subculture was obtained by adding 100 μL of overnight subculture in 10 mL MHB and incubated for a further 2−3 h. Finally, the $OD_{600}$ value of the bacterial suspension was determined and the colony forming units per millilitre (CFU mL$^{-1}$) extrapolated. A full dilution of the bacterial cell suspension in saline was achieved by washing (3220 × g for 10 min) and the $OD_{600}$ in saline determined. The suspension was then diluted out accordingly in order to achieve a $2 \times 10^6$ CFU mL$^{-1}$ solution ($10^5$ CFU in 50 μL thigh$^{-1}$).

**Quantification of injected MRSA solution:** In order to be able to correlate the actual CFU mL$^{-1}$ present in the MSRA injection solution with the estimated CFU mL$^{-1}$ based on the $OD_{600}$ readings, a standard plate count from the MSRA injection solution was performed. Thus, 10 μL of the injectable MRSA suspension was diluted down to 10–1000 fold, each dilution plated out onto agar and incubated at 37 °C for 24 h. From the estimated $2 \times 10^6$ CFU mL$^{-1}$ solution, 18 CFU per 10 μL were found in the 1:1000 dilution, giving a final concentration of $1.8 \times 10^6$ CFU mL$^{-1}$ for the actual injectable MRSA solution.

**In vivo experimental assay:** Eight-week-old female outbred CD1 mice (UQBR-AIBN) were rendered neutropenic by injecting two doses of cyclophosphamide intraperitoneally 4 days (150 mg kg$^{-1}$) and 1 day (100 mg kg$^{-1}$) prior to experimental infection. The infection model using MRSA was established by intramuscular injection of 50 μL of early-log-phase bacterial MRSA suspension (around $2 \times 10^6$ CFU mL$^{-1}$) in saline into both thigh muscles. After 2 h, a single dose of vancomycin (200 mg kg$^{-1}$) or compound of this invention (25 mg kg$^{-1}$) was administered by a subcutaneous injection over the interscapular (area at back of the neck). Untreated animals received equivalent volume of saline (Baxter). The mice were monitored for signs of normal behaviour (i.e. grooming, eating, drinking, sleeping and alertness) during and following dosing. Two h after saline/antibiotic treatment, 0.05 mL of blood was collected by tail incision. 24 h after MRSA infection, mice were killed and blood collected from the heart by cardiac puncture (saline group) or by tail incision (vancomycin and vancapticin treated groups). For each mouse, both thighs were collected aseptically by cutting the leg at the hip and knee, placed in 10 mL of cold sterile saline and the individual weight of each thigh recorded.

**Plasma sample preparation:** Blood samples were taken at 2 h (tail incision) and 22 h (cardiac puncture or tail incision) post saline/vancomycin/vancapticin treatment using Lithium-Heparin Microvette® (Sardest) or using heparin coated syringes. All samples were kept at 4 °C and spun down at 10,000 × g for 15 min. Plasma was collected and kept at −80 °C until used.

**Thigh homogenates and CFU determination:** Thighs were homogenised at 20,000 rpm for 15 s using a Polytron MR2500E using a 200 mm probe (Kinematica). Homogenate solutions were filtered using a 100 μm pore size filter (BD) and 1 mL of filtrate solution placed on ice and serial dilutions promptly done (1:10 and 1:100) and seeded onto appropriate nutrient agar plates (Bactolaboratories) and incubated at 37 °C overnight. Colonies were counted the next day and CFU thigh$^{-1}$ and the CFU g$^{-1}$ of thigh calculated based on the plate count and dilution factor.

***S. pneumoniae* lung infection in vivo efficacy model.** Performed by Eurofins, Taiwan, following similar previous literature procedures[29, 53].

**Summary:** The study objective was to evaluate the antimicrobial activity of test articles in a lung infection model with *S. pneumoniae* (ATCC$^\circ$ 6301™). Male ICR mice were inoculated intratracheally (IT) with *S. pneumoniae* at an inoculum size of $2.96 \times 10^6$ CFU per mouse (20 μL per mouse). Vancapticins **19** and **24** were administered at 25 mg kg$^{-1}$ to separate groups of mice by subcutaneous (SC) injection 2 h post-infection. The reference standard, vancomycin **1**, was dosed at 25 mg kg$^{-1}$ qd SC at 2 h after IT inoculation. Mortality was observed for 10 days. An increase in survival rate of 50 percent or more ( ≥ 50%) of the treated animals relative to the vehicle group indicates significant antimicrobial activity. Subcutaneous injection (SC) of **19** and **24** at 25 mg kg$^{-1}$ was associated with pronounced increase in the 10-day survival rate relative to the vehicle group, indicating significant antimicrobial activity. The vancomycin treated group, 25 mg kg$^{-1}$ SC qd, also exhibited significant antimicrobial activity ( ≥ 50% survival rate) relative to the vehicle control.

**Protocol:** Test substances at 25 mg kg$^{-1}$ were each administered subcutaneously (SC) to test animals 2 h after inoculation. The dosing volume was 10 mL kg$^{-1}$.

**Animals:.** Male ICR mice weighing 22 ± 2 g were provided by BioLasco Taiwan (under a Charles River Laboratories Technology Licensee). Space allocation for 5 animals was 29 × 18 × 13 cm. Mice were housed in animal cages and were maintained in a controlled temperature (20–23 °C) and humidity (50–80%)

environment with 12 h light/dark cycles for at least three days at Eurofins Panlabs Taiwan, Ltd. laboratory prior to use. Free access to standard lab chow for mice (Laboratory Rodent Diet MFG (Oriental Yeast Co., Ltd. Japan)) and tap water in bottle were granted. All aspects of this work including housing, experimentation and disposal of animals were performed in general accordance with the Guide for the Care and Use of Laboratory Animals (National Academy Press, Washington, DC, 2011).

**Organism:** *S. pneumoniae* (ATCC$^\circ$ 6301™) was obtained from the American Type Culture Collection (Rockville, MD, USA). The bacterial culture was diluted in brain–heart infusion broth containing to achieve the target inoculum size of $1.0–5.0 \times 10^6$ CFU per 20 μL that would be inoculated intratracheally into each animal.

**Chemicals:** Bacto agar (Cat# 214040, BD DIFCO, USA), brain heart infusion broth (Cat# 237500, BD, USA), fetal bovine serum (Cat# 30071.03, HyClone, USA), sodium chloride (Cat# S7653, SIGMA-Aldrich, USA), vancomycin (Cat# V2002, Sigma, USA) and water for injection (WFI) (Tai-Yu, Taiwan).

**Equipment:** Animal cage (Allentown, USA), biokinetic reader (Bio-Tek, USA), biological safety cabinet (NuAire, USA), centrifuge (Eppendrof 5810 R, Germany), laminar flow (Chao-Shin, Taiwan), orbital shaking incubator (Firstek Scientific, Taiwan), Pipetman P100 (Gilson, France), refrigerated incubator (Hotpack, USA) and ultra-low temperature freezer (NuAire, USA).

**Methods:** Assay # 608100 *S. pneumoniae* (ATCC$^\circ$ 6301™), Infected Lung Model. Groups of 10 male specific-pathogen-free ICR mice weighing 22 ± 2 g were used. Acute pneumonia was induced by intratracheal inoculation with a $LD_{90–100}$ dose ($2.96 \times 10^6$ CFU mouse$^{-1}$) of *S. pneumoniae* (ATCC$^\circ$ 6301™) suspended in 20 μL of BHI. Vehicle (10 mL kg$^{-1}$), vancomycin and test substances at 25 mg kg$^{-1}$ were each administered subcutaneously 2 h post-infection. Mortality was recorded daily for 10 days following inoculation. Increase of 50 percent or more ( ≥ 50%) in survival rate relative to the vehicle control group indicates significant anti-infective activity.

**Bioluminescence IP infection in vivo model.** Performed at the University of Queensland, following similar previous literature procedures[27]. Xen-29 (Caliper LifeSciences), a methicillin-sensitive *S. aureus* (MSSA) strain which contains a copy of the modified *luxABCDE* operon of *Photorhabdus luminescens* was taken from the storage at −80 °C and freshly seeded on agar plates for overnight growth at 37 °C. From the overnight culture preparation, a single colony (tested for bioluminescence) was diluted into 10 mL of MHB and incubated overnight at 37 °C. A mid log-phase subculture was obtained by adding 100 μL of overnight subculture in 10 mL MHB and incubated for a further 2–3 h. Finally, the $OD_{600}$ value of the bacterial suspension was determined and the colony forming units per millilitre (CFU mL$^{-1}$) was extrapolated. The suspension was then diluted out accordingly in order to achieve a $2.5 \times 10^8$ CFU mL$^{-1}$ solution ($2.5 \times 10^7$ CFU in 100 μL). Seven-week-old female outbred CD1 mice (UQBR-UQCCR) were rendered neutropenic by injecting two doses of cyclophosphamide (Sigma-Aldrich) intraperitoneally 4 days (150 mg kg$^{-1}$) and 1 day (100 mg kg$^{-1}$) prior to experimental infection. The peritonitis infection model using bioluminescent Xen-29 was established by intraperitoneal injection of 100 μL of the adjusted bacterial Xen-29 suspension (around $2.5 \times 10^7$ CFU) in saline. 30 min after bacterial inoculation, a single dose of daptomycin (50 mg kg$^{-1}$), vancomycin (200 mg kg$^{-1}$) or vancapticin (25 mg kg$^{-1}$) was administered by a subcutaneous injection over the interscapular area (area at back of the neck). Untreated animals received equivalent volume of saline (Baxter). Mice were monitored for signs of normal behaviour (i.e. grooming, eating, drinking, sleeping and alertness) during and following dosing. Blood samples (50 μL) were taken 5.5 h post Xen-29 inoculation by tail bleed using Lithium-Heparin Microvettes® (Sardest). All samples were kept at 4 °C and spun down at 10,000 × g for 15 min. Plasma samples were collected and kept at −80 °C until used; while the pellets were re-suspended in 100 μL of sterile saline, serial diluted and spread on agar plates for overnight incubation at 37 °C. Xen-29 colonies were then counted, with the CFU mL$^{-1}$ blood calculated. After killing, each mouse was opened and intraperitoneal swabs were taken for CFU determination. Individual spleens were also harvested aseptically, homogenised in 100 μL of sterile saline by crushing the tissue with a syringe plunger against a 50 μm filter, serial diluted and plated on agar plates for O/N incubation at 37 °C. Xen-29 colonies from intraperitoneal swabs and spleens were then verified. Luminescent images were acquired using the Xenogen IVIS-200 Optical In Vivo Imaging System (PerkinElmer). Luminescent images were quantified with the IVIS Living Image software where the total flux (number of photons/second) was calculated by a user defined region of interest (ROI) covering the infection sites.

**Data availability.** All relevant data are available in this article and its Supplementary Information files, or from the corresponding authors on request.

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

## Acknowledgements

We thank Caliper Biosciences (Perkin Elmer) for supply of the bioluminescent MSSA bacteria Xen-29. M.A.C. is a current National Health and Medical Research Council (NHMRC) Principal Research Fellow (APP1059354) and former Australia Fellow (AF511105). J.A.R. is a NHMRC Career Development Fellow (APP1048652). This work was supported by a Wellcome Trust Seeding Drug Discovery Award 094977/Z/10/Z, and NHMRC Project Grants APP631632 and APP1026922. R.A.G.S acknowledges the support of the UK Medical Research Council and the UK NIHR Biomedical Research Council at Guy's Hospital, London. The following strains were provided by the Network on Antimicrobial Resistance in *S. aureus* (NARSA) for distribution by BEI Resources, NIAID, NIH: *S. aureus* NRS-1, and VRS-4. Strains *S. aureus* ATCC 4300, ATCC 29213, *S. pneumoniae* ATCC 700677 and *E. faecium* ATCC 51559, *B. megaterium* KM (ATCC 13632), *B. subtilis* NRS-231 (ATCC 6633) and human cell lines HepG2 (ATCC HB-8065) and HEK293 (ATCC CRL-1573), were acquired from American Type Culture Collection (ATCC).

## Author contributions

M.A.C. designed the project with input from J.R.B., A.P.G.B., R.A.G.S. (design of the electrostatic effector sequence), and M.A.T.B., J.A.R. and D.L.P. The manuscript was written by M.A.T.B., M.A.C. and K.A.H. with input from all authors. M.A.T.B., M.A.C. and M.S.B. designed and coordinated experiments and analysed results. Y.G., J.R.B., C.M., R.P., T.A.B., T.K., K.A.H., Z.Z., F.L. J.C.L. and J.Z. designed, synthesised, purified and analysed compounds. M.S.B., R.P. and D.J.E. purified and analysed compounds. Y.G., So.R., A.M.K., M.A., A.G.E., W.P., M.C., J.X.H., Z.Z., J.E.D., N.H.D., Se.R., A.B.S. and H.E.S. carried out in vitro and in vivo biological assays. All authors discussed the results and commented on the manuscript.

## Additional information

**Competing interests:** M.A.C. and M.A.T.B are listed as inventors in patent application WO 2015/117196-A1 'Antibacterial Agents', which includes compounds described in this article. The remaining authors declare no competing financial interests.

