## [Peer Review File · Nature Communications]

Reviewers' comments:

Reviewer #1 (Remarks to the Author):

The paper "Protein inspired antibiotics active against vancomycin and daptomycin resistant bacteria" presents a novel class of compounds, the 'vancaptins' that are formed from peptide libraries ligated to N-terminal lipophilic membrane insertive elements that were then ligated to vancomycin via chemical linkers. The resulting 'vancaptins' have enhanced membrane affinity and increase the local drug concentration at the target site. These molecules indeed have high levels of activity, with the best compounds 20- to 100- fold more potent than vancomycin or daptomycin by MIC against a range of Gram positive bacteria, including vancomycin resistant strains.

The three compounds examined more fully after MIC determination are 16, 18, and 23, all of which have Lys based linkers. Compounds 18 and 23 both demonstrated potent bactericidal effects against MRSA in neutropenic mice. Similarly testing both compounds in a *S. pneumoniae* model and in an intra-peritoneal infection model showed good efficacy, particularly for compound 18.

Since the 'vancaptins' are composed of vancomycin as well as the MIE, EEPS, and linker, the authors next sought to determine if the compounds act via the same MOA as vancomycin, or if the additional motifs leads to a novel mechanism of action. Compounds 16 and 23 inhibited peptidoglycan formation in a cell free radiolabeled assay, and required over 10,000-fold molar excess of competitive inhibitor, compared to 100-fold molar excess for vancomycin, to neutralize antibiotic activity. Further by ITC the authors demonstrate that increased dimerization is not occurring, and thus is not responsible for the increased potency.

SPR studies support the notion that the 'vancaptins' have increased association with membranes, with stronger binding observed with anionic membranes, and membranes derived from MRSA over *E. coli* or mammalian cells. Although the authors report minimal damage to the cytoplasmic membrane of *S. aureus* as measured by cytoplasmic leakage, there is an increase of cytoplasmic leakage of compound 23 in particular compared to vancomycin, a point that should be addressed in the manuscript. Transmission electron microscopy of *S. aureus* treated with compounds 16 and 23 at 16x their MICs demonstrated greater structural changes than treatment with vancomycin, suggesting an increase in antibiotic potency.

This work presents excellent evidence that the novel 'vancaptins' have significantly more potent MICs against multiple Gram-positive bacteria, including MRSA, MDR *S. pneumoniae*, and other vancomycin resistant strains. There is excellent biochemical and in vitro experimentation that characterized the MOA of these compounds, however the authors have not clearly demonstrated that the mode of action (MOA) is the same as vancomycin, or if the added derivations have altered the MOA in living cells. The great majority of MOA studies in this work are performed in vitro and this work would be greatly strengthened if an in vivo demonstration of MOA were included.

P2, line 60: After a description of the structure of the derivatives of vancomycin they refer to Figure 1, which does not correspond with the description in the text. Do the authors mean to refer to table 1 (which includes an image of the modified vancomycin)?

Figure 1, line 247: Labeling is incorrect in the figure legend, as b. in the figure legend describes c. in the figure and c. in the figure legend describes b. in the figure. There is no description of panel e. besides a title.

Reviewer #2 (Remarks to the Author):

This reviewer would like to congratulate the authors on a very nicely composed and thoroughly researched manuscript. The authors present a novel and rationally designed series of vancomycin analogs aimed at addressing the rising incidence of drug-resistant Gram-positive pathogens and specifically, vancomycin resistant variants. Utilizing microbiological techniques the authors convincingly demonstrate structure activity relationships with respect to antimicrobial activity and show that the designed molecules are potent inhibitors of bacterial growth, including drug resistant variants. The authors are cautious to consider desired drug properties and in so doing develop a second series to successfully address pharmacokinetics and ADME. Optimized molecules are then assayed for in vivo translation in various models of animal infection. These studies are well designed and incorporate highly relevant controls. The novel molecules exhibit very potent efficacy. Finally, the authors do a wonderful job investigating and ultimately elucidating the mode(s) of action of representative examples of the novel molecules. The authors have appropriately cited the relevant literature and nicely acknowledged other advancements in the field.

In short, this manuscript describes a series of novel Gram-positive antimicrobial agents with extensive supporting data. The findings presented in this manuscript should be of high interest to those in the field of antimicrobial drug discovery and given the threat posed by drug resistant pathogens, the community at large! This reviewer overwhelmingly supports publication of the manuscript without modification.

Reviewer #3 (Remarks to the Author):

This paper elegantly demonstrates the in vitro and in vivo activity of a series of vancomycin derivatives tailored to interact with the bacterial membrane. It also brings some pieces of information regarding their mode of action and their specificity of action towards Gram-positive bacteria. In an era of increasing resistance, the series of compounds presented is of interest for a broad readership.

- a) The data are globally convincing, but the paper needs to be carefully re-read by the authors. The reading is not easy, as the paper is extremely short. The writing is very concise and the reader is referred to the supplementary material all the time. Moreover, there are some mistakes that complicate the reading. To highlight some of them: Figure 1, panel b is panel c, and vice-versa; the Y axis of panel d is labeled "MRSA", but the caption refers to a MSSA bioluminescent strain, panel b is not is % but a ratio; line 162: comparison is between 23 and 18, I guess; table S7: I guess the first series of data is for vancomycin). Some acronyms are not defined (even in the abbreviation list (HS, for example)). In my view, defining abbreviations in Tables and Figures makes things much easier than forcing the reader to refer to the abbreviation list all the time. I leave however this decision to the appreciation of the editor.
- b) The number of replicates and of independent experiments should be systematically mentioned in each figure/table, together with the statistical analysis results (including the test performed). Also, when pertinent, the concentrations in X MIC or specific conditions should be added in the figure/table legend to make reading easier.
- c) It is regrettable that oritavancin has not been included in the study. This lipoglycopeptide is indeed the most active one against VRSA and VRE among the currently marketed molecules.
- d) Susceptibility testing: CLSI recommend adding 0.002 % polysorbate when determining susceptibility to lipoglycopeptides, which does not seem to have been done here (the authors state

they used non-binding surface plates). As this is not the standard procedure, it makes difficult comparison with literature data. If no polysorbate has been added in the assay (throughout the paper) it may have a negative influence on the activity of the comparators. By the same token, polysorbate could also improve the activity of the described compounds; has this been tested? Likewise, MBC were determined making use of resazurin metabolisation assay. Again, this is not a standard procedure. What is the limit of detection for this assay (in other words, the minimal metabolically active inoculum to generate a change in coloration?). Moreover, I do not see a description of these MBCs in the manuscript. MICs are compared to those of vancomycin in the text; they should also be compared to those of the other control antibiotics.

e) Lines 113-116: the authors mention that the concentration of their compound remains above the MIC for at least 12 h. This is not useful per se, depending of the pharmacodynamics profile of the drug. A bactericidal antibiotic is most probably AUC- or C_{max}-dependent rather than time-dependent. The paper does not present enough data to determine which PD parameter is of importance here, but insisting on time above MIC is probably meaningless.

f) Figure 2c does not illustrate a competitive ligand antagonism assay.

g) In figure 1, panel B, the scale selected does not allow to see differences among treatments. The caption describes some subtle changes in bioluminescence signal which are not at all visible in the figure.

h) The weakest part of the paper is the study of the mechanism of action/interaction with membranes.

a. Membrane binding is globally similar towards eukaryotic and prokaryotic membranes. In these conditions, what does explain the specificity of action?

b. It could have been interesting to perform some mechanistic studies on strains resistant to vanco/dapto in order to check whether these mechanism of resistance also modify the interaction of the compounds with the bacteria.

c. Membrane permeabilization is low also for daptomycin. This is surprising, as this is claimed to be its mode of action.

d. Electron microscopy images show very different changes induced by compounds 16 and 23. This is not commented in the text. This suggests a different mode of interaction with the membranes, which has not been studied. The MIE is different in these two compounds and therefore the interaction with the membrane is also probably different.

i) In table S3, there is an apparent increase in the concentration of vancomycin and compound 16 over the first hours of incubation. What is the interpretation of the authors?

Reviewers' comments:

Reviewer #1 (Remarks to the Author):

The paper "Protein inspired antibiotics active against vancomycin and daptomycin resistant bacteria" presents a novel class of compounds, the 'vancapticins' that are formed from peptide libraries ligated to N-terminal lipophilic membrane insertive elements that were then ligated to vancomycin via chemical linkers. The resulting 'vancapticins' have enhanced membrane affinity and increase the local drug concentration at the target site. These molecules indeed have high levels of activity, with the best compounds 20- to 100- fold more potent than vancomycin or daptomycin by MIC against a range of Gram positive bacteria, including vancomycin resistant strains.

The three compounds examined more fully after MIC determination are 16, 18, and 23, all of which have Lys based linkers. Compounds 18 and 23 both demonstrated potent bactericidal effects against MRSA in neutropenic mice. Similarly testing both compounds in a *S. pneumoniae* model and in an intra-peritoneal infection model showed good efficacy, particularly for compound 18.

Since the 'vancapticins' are composed of vancomycin as well as the MIE, EEPS, and linker, the authors next sought to determine if the compounds act via the same MOA as vancomycin, or if the additional motifs leads to a novel mechanism of action. Compounds 16 and 23 inhibited peptidoglycan formation in a cell free radiolabeled assay, and required over 10,000-fold molar excess of competitive inhibitor, compared to 100-fold molar excess for vancomycin, to neutralize antibiotic activity. Further by ITC the authors demonstrate that increased dimerization is not occurring, and thus is not responsible for the increased potency.

SPR studies support the notion that the 'vancapticins' have increased association with membranes, with stronger binding observed with anionic membranes, and membranes derived from MRSA over *E. coli* or mammalian cells. Although the authors report minimal damage to the cytoplasmic membrane of *S. aureus* as measured by cytoplasmic leakage, there is an increase of cytoplasmic leakage of compound 23 in particular compared to vancomycin, a point that should be addressed in the manuscript. Transmission electron microscopy of *S. aureus* treated with compounds 16 and 23 at 16x their MICs demonstrated greater structural changes than treatment with vancomycin, suggesting an increase in antibiotic potency.

We repeated the diSC3(5) assay (Fig. 7) with fresh compound stocks as the discrepancy between compound 16 and 23 (now labeled 17 and 24, respectively) seemed suspicious. Furthermore, we felt that using citropin as a positive control, and reporting data relative to this control, was potentially misleading since any effect induced by the glycopeptides might appear inconsequential

by comparison. At the time, we did not have access to oritavancin, which was used as a positive control in the current study. In Fig 7c we now report the data as a function of absolute fluorescence intensity, which demonstrated that the three vancapticin analogues induced membrane depolarization in a similar manner to oritavancin. Under the conditions of the assay we did not observe any effect for daptomycin, vancomycin or dalbavancin. To further assess membrane effects, we also examined permeabilisation effects using propidium iodide (Fig 7d). Here, compounds 17 and 24, but not 19, exerted an effect similar to oritavancin, whereas daptomycin induced the greatest comparative level of membrane permeabilisation.

This work presents excellent evidence that the novel 'vancapticins' have significantly more potent MICs against multiple Gram-positive bacteria, including MRSA, MDR *S. pneumoniae*, and other vancomycin resistant strains. There is excellent biochemical and in vitro experimentation that characterized the MOA of these compounds, however the authors have not clearly demonstrated that the mode of action (MOA) is the same as vancomycin, or if the added derivations have altered the MOA in living cells. The great majority of MOA studies in this work are performed in vitro and this work would be greatly strengthened if an in vivo demonstration of MOA were included.

As described above, we conducted additional assays using fluorescent probes to assess membrane activity of key compounds in 'in vivo' assays against MRSA (and MSSA), based on our interpretation of the reviewer's suggestion for 'in vivo' MoA. The results clearly showed that the vancapticins behave more like oritavancin than vancomycin, providing strong support for the proposed MoA in whole cells.

P2, line 60: After a description of the structure of the derivatives of vancomycin they refer to Figure 1, which does not correspond with the description in the text. Do the authors mean to refer to table 1 (which includes an image of the modified vancomycin)?

Yes – this has been corrected by separating the figure from the table, and renumbering the figures.

Figure 1, line 247: Labeling is incorrect in the figure legend, as b. in the figure legend describes c. in the figure and c. in the figure legend describes b. in the figure. There is no description of panel e. besides a title.

Now corrected.

Reviewer #2 (Remarks to the Author):

This reviewer would like to congratulate the authors on a very nicely composed and thoroughly researched manuscript. The authors present a novel and rationally designed series of vancomycin analogs aimed at addressing the rising incidence of drug-resistant Gram-positive pathogens and specifically, vancomycin resistant variants. Utilizing microbiological techniques the authors convincingly demonstrate structure activity relationships with respect to antimicrobial activity and show that the designed molecules are potent inhibitors of bacterial growth, including drug resistant variants. The authors are cautious to consider desired drug properties and in so doing develop a second series to successfully address pharmacokinetics and ADME. Optimized molecules are then assayed for in vivo translation in various models of animal infection. These studies are well designed and incorporate highly relevant controls. The novel molecules exhibit very potent efficacy. Finally, the authors do a wonderful job investigating and ultimately elucidating the

mode(s) of action of representative examples of the novel molecules. The authors have appropriately cited the relevant literature and nicely acknowledged other advancements in the field.

In short, this manuscript describes a series of novel Gram-positive antimicrobial agents with extensive supporting data. The findings presented in this manuscript should be of high interest to those in the field of antimicrobial drug discovery and given the threat posed by drug resistant pathogens, the community at large! This reviewer overwhelmingly supports publication of the manuscript without modification.

No points to address – reviewer supports publication.

Reviewer #3 (Remarks to the Author):

This paper elegantly demonstrates the in vitro and in vivo activity of a series of vancomycin derivatives tailored to interact with the bacterial membrane. It also brings some pieces of information regarding their mode of action and their specificity of action towards Gram-positive bacteria. In an era of increasing resistance, the series of compounds presented is of interest for a broad readership.

a) The data are globally convincing, but the paper needs to be carefully re-read by the authors. The reading is not easy, as the paper is extremely short. The writing is very concise and the reader is referred to the supplementary material all the time.

We thank the reviewer for this feedback and agree that expansion of key areas would lend greater readability. As such, we have moved a substantial amount of material relevant to the core narrative of the paper from the supplementary section to the main manuscript, in order to avoid the need to refer to this section as often. We have also re-written and expanded some sections of writing to aid in clarification.

Moreover, there are some mistakes that complicate the reading. To highlight some of them: Figure 1, panel b is panel c, and vice-versa; the Y axis of panel d is labeled “MRSA”, but the caption refers to a MSSA bioluminescent strain, panel b is not % but a ratio; line 162: comparison is between 23 and 18, I guess; table S7: I guess the first series of data is for vancomycin).

All now corrected.

Some acronyms are not defined (even in the abbreviation list (HS, for example)). In my view, defining abbreviations in Tables and Figures makes things much easier than forcing the reader to refer to the abbreviation list all the time. I leave however this decision to the appreciation of the editor.

Acronyms are now defined on first use within the text, and as footnotes within tables and figures.

b) The number of replicates and of independent experiments should be systematically mentioned in each figure/table, together with the statistical analysis results (including the test performed). Also, when pertinent, the concentrations in X MIC or specific conditions should be added in the figure/table legend to make reading easier.

All items corrected.

c) It is regrettable that oritavancin has not been included in the study. This lipoglycopeptide is indeed the most active one against VRSA and VRE among the currently marketed molecules.

At the time these experiments were conducted oritavancin was not commercially available; nor was the precursor (chloroeremomycin) that would allow for its synthesis (dalbavancin was also not available, but we were able to synthesise it internally in our labs). We have since procured oritavancin, and have now included it as a comparator for MIC and membrane activity mode of action studies as requested.

d) Susceptibility testing: CLSI recommend adding 0.002 % polysorbate when determining susceptibility to lipoglycopeptides, which does not seem to have been done here (the authors state they used non-binding surface plates). As this is not the standard procedure, it makes difficult comparison with literature data. If no polysorbate has been added in the assay (throughout the paper) it may have a negative influence on the activity of the comparators. By the same token, polysorbate could also improve the activity of the described compounds; has this been tested ?

We have conducted extensive comparisons of MIC testing in different types of plates and compared the results to the use of additives, which is the subject of a manuscript in preparation. Specifically, for lipoglycopeptides such as dalbavancin, oritavancin, and the vancaptins, the use of NBS plates gives results identical to the use of 0.002 % polysorbate, showing substantial improvements in activity compared to MICs determined in untreated polystyrene with no additive. This is now described in detail in the methods section.

Likewise, MBC were determined making use of resazurin metabolisation assay. Again, this is not a standard procedure. What is the limit of detection for this assay (in other words, the minimal metabolically active inoculum to generate a change in coloration ?). Moreover, I do not see a description of these MBCs in the manuscript.

The MBC determination protocol has been removed from the methods as it indeed was not mentioned in the manuscript due to its non-standard nature.

MICs are compared to those of vancomycin in the text; they should also be compared to those of the other control antibiotics.

MIC comparisons to daptomycin, telavancin, dalbavancin and oritavancin have been added, as have efficacy comparisons.

e) Lines 113-116: the authors mention that the concentration of their compound remains above the MIC for at least 12 h. This is not useful per se, depending of the pharmacodynamics profile of the drug. A bactericidal antibiotic is most probably AUC- or Cmax-dependent rather than time-dependent. The paper does not present enough data to determine which PD parameter is of importance here, but insisting on time above MIC is probably meaningless.

Historically, for many antibiotics, time above MIC has been used as an initial surrogate for efficacy until more accurate testing can determine the PK/PD driver. While vancomycin is now generally accepted as fAUC/MIC, historically $t_{T>MIC}$ was initially reported to be a better predictor of the

antibacterial effect (Antimicrob Agents Chemother. 2011 Oct; 55(10): 4619–4630. doi: 10.1128/AAC.00182-11).

f) Figure 2c does not illustrate a competitive ligand antagonism assay.

This figure (now 6c) does show how addition of free Ac-Lys(Ac)-D-Ala-D-Ala ligand can antagonize (compete) for binding to antibiotic compared to membrane bound native -Lys(Ac)-D-Ala-D-Ala (or for that matter L-DAP-D-Ala-D-Ala) native ligands in bacteria. Dose-dependent addition of synthetic ligand antagonised antibiotic activity against whole bacteria. We would be happy to use an alternative term if the reviewer would like to proffer one, however we feel most scientists would understand the term used in the paper.

g) In figure 1, panel B, the scale selected does not allow to see differences among treatments. The caption describes some subtle changes in bioluminescence signal which are not at all visible in the figure.

The scale has been adjusted.

h) The weakest part of the paper is the study of the mechanism of action/interaction with membranes. a. Membrane binding is globally similar towards eukaryotic and prokaryotic membranes. In these conditions, what does explain the specificity of action ?

The SPR membrane binding data contained in the previous version was somewhat confusing and conflicting – we have removed the data using membrane preparations and focused on the results from model membranes, which provide clear evidence of increased binding to negatively charged membrane surfaces.

b. It could have been interesting to perform some mechanistic studies on strains resistant to vanco/dapto in order to check whether these mechanism of resistance also modify the interaction of the compounds with the bacteria.

We agree, but unfortunately our internal biosafety risk assessments place restrictions on the use of required instrumentation for the mechanistic studies, so we are unable to analyse the potentially more hazardous vancomycin/daptomycin-resistant strains.

c. Membrane permeabilization is low also for daptomycin. This is surprising, as this is claimed to be its mode of action.

As described above, we have repeated these membrane assays under different conditions, and with different types of membrane dyes. The assays are highly sensitive to the timing of dye and compound addition, and length of compound exposure. With Figure 7c, we now clearly see that daptomycin causes membrane permeabilisation, though in Figure 7d (and S4) there is no depolarisation observed. However, the depolarisation results match literature data for oritavancin (Antimicrob Agents Chemother. 2009 Mar; 53(3): 918-925. doi: 10.1128/AAC.00766-08), giving confidence that the procedure is working. The same trend was observed using both MRSA and MSSA strains.

d. Electron microscopy images show very different changes induced by compounds 16 and 23. This is not commented in the text. This suggests a different mode of interaction with the membranes,

which has not been studied. The MIE is different in these two compounds and therefore the interaction with the membrane is also probably different.

The SEM images showed variability between individual cells, and it was difficult to select a small number of representative images that accurately reflected the overall types of variability seen. However, in general the vancapticins showed greater cell damage than vancomycin, which in turn showed cell alterations compared to untreated cells. Given the unavoidable bias in selecting which cell images to present to represent the general situation, we decided it was better to remove these figures altogether.

i) In table S3, there is an apparent increase in the concentration of vancomycin and compound 16 over the first hours of incubation. What is the interpretation of the authors ?

The inconsistent results from these plasma stability assays is attributable to considerable variability through the assay process. The assay has been optimized and repeated, particularly with respect to the order of addition of reagents during the plasma sample preparations, giving more consistent results.

Additional manuscript reformatting:

- 1) reduction of abstract from >300 words with references to <150 words with exclusion of references.
- 2) movement of Tables 2 and 3 and Figures 2, 3, and 4 from the Supplementary information to the main manuscript. A large portion of the detailed methods have also been moved, replacing the short methods summary in the previous version. Detailed chemical characterization has been left in the SI. Manuscript text, excluding Methods, is 2500 words.
- 3) Re-numbering of all compounds to account for the inclusion of oritavancin.

Reviewers' comments:

Reviewer #3 (Remarks to the Author):

I thank the authors for their replies and the changes introduced in the manuscript. This version has been improved but I have still some difficulties essentially regarding the mechanistic studies.

1. Permeability studies: how do the authors explain that the effect of daptomycin is so high as compared to that of oritavancin, despite the fact that the concentration used for oritavancin represents much higher multiples of the MIC than for daptomycin? I could not find a reference in which both products were tested in parallel using the same protocol but, based on these data, I would suspect an artefact increasing the fluorescence signal in the presence of dapto (some papers suggest an interaction of PI with calcium ions, for example) or decreasing it in the presence of the other agents.

Based on these data, the authors suggest that vancomycins have a mode of action similar to that of oritavancin. However, vancomycins are not very rapidly bactericidal (6 h for a 5 log reduction in inoculum in Figure 2), while oritavancin causes a 7 log decrease in 30 minutes in the papers of Belley et al., indicating a much more rapid cidal effect. Thus, these data are not coherent with the fact that the two types of drugs seem to cause in the present study similar effects on the membrane.

Thus, this part remains weak in my view: link between depolarization permeabilization, and killing.

2. All the mechanistic studies are presented in the discussion, which is very unusual. The description of the data should be in the results section, and the discussion should consider the data as a whole to give an interpretation. As it is now, data discussion is rather limited.

3. Caption to figure 6 is very obscure. Panel c is a model and I do not see how it illustrates the antagonism by free ligand. This brings me back to my comment to the original submission. This panel is a model for a possible interaction but NOT a ligand-antagonism ASSAY, which by definition, should correspond to experimental data. I apologize if I was unclear.

4. Minor comment: the source of new products has not been added in the material section, if I am correct.

REPLY TO REVIEWER COMMENTS

NCOMMS-16-27949-T: Protein inspired antibiotics active against vancomycin and daptomycin resistant bacteria

Reviewer 3

1. *Permeability studies: how do the authors explain that the effect of daptomycin is so high as compared to that of oritavancin, despite the fact that the concentration used for oritavancin represents much higher multiples of the MIC than for daptomycin ? I could not find a reference in which both products were tested in parallel using the same protocol but, based on these data, I would suspect an artefact increasing the fluorescence signal in the presence of dapto (some papers suggest an interaction of PI with calcium ions, for example) or decreasing it in the presence of the other agents.*

The assay conditions used for the permeability assays are done **without** added polysorbate-80 to avoid possible membrane permeabilisation caused by this nonionic surfactant, or fluorescence interference as previously reported (Belley, AAC 2009 53, 918). We also found that NBS plates we use for MIC assays interfered with the assay, so employed standard polystyrene plates. The MIC in the absence of polysorbate, in standard (non-NBS) polystyrene plates used for the membrane assays, are $1 \mu\text{g mL}^{-1}$ for daptomycin, $1 \mu\text{g mL}^{-1}$ for oritavancin, $0.25 \mu\text{g mL}^{-1}$ for dalbavancin, $1 \mu\text{g mL}^{-1}$ for **17**, and $2 \mu\text{g mL}^{-1}$ for **24**. This MIC data has now been added as Supplementary Table S10. Therefore the concentrations used for daptomycin and oritavancin in the permeability assays ($16 \mu\text{g mL}^{-1}$) are at equivalent 16-fold levels above their MIC. The approximately 6-fold enhanced permeability observed for daptomycin is presumably due to this property representing its main mechanism of action, in contrast to oritavancin's multiple modes.

Based on these data, the authors suggest that vancapticins have a mode of action similar to that of oritavancin. However, vancapticins are not very rapidly bactericidal (6 h for a 5 log reduction in inoculum in Figure 2), while oritavancin causes a 7 log decrease in 30 minutes in the papers of Belley et al., indicating a much more rapid cidal effect. Thus, these data are not coherent with the fact that the two types of drugs seem to cause in the present study similar effects on the membrane.

It is difficult to directly compare the time kill results, as they are conducted under different conditions. In one Belley paper (AAC 2009 53, 918, Figure 1B) oritavancin shows approximately 4 log reduction from initial inoculum (7.5 to 3.5 log CFU/mL) in 3h for a concentration of $16 \mu\text{g mL}^{-1}$; with $4 \mu\text{g mL}^{-1}$ there is a 3 log reduction. These experiments were conducted in nutrient-depleted media, which would be expected to make the bacteria more susceptible to killing than the MHB broth used in our time kill assays. No polysorbate-80 additive was used. These results are not that dissimilar to Cmpd 24, which at $0.25 \mu\text{g mL}^{-1}$ induced a 3.8 log reduction in 6h.

More rapid rates of killing, as indicated by the reviewer, are seen in more recent papers (Mckay, J Antimicrob Chemother 2009, 63 1191: Fig 1A 4.5 log decrease in 1h at $4 \mu\text{g mL}^{-1}$; 4; Belley AAC 2016 60 4342: Fig 1A 4.5 log decrease in 30 min at $16 \mu\text{g mL}^{-1}$, 4.5 log decrease in 1h at $4 \mu\text{g mL}^{-1}$). Both of these assays are conducted in CAMHB with added polysorbate-80, which McKay indicates has a pronounced effect on time-kill outcome.

Our time kill assays were conducted in NBS plates, with a lower absolute concentration of compound than the oritavancin papers, though with a higher-fold excess relative to MIC. If the NBS assay conditions correspond to the use of added polysorbate-80 in a time kill assay, then indeed oritavancin shows faster killing. We do appear to have faster time kill than dalbavancin (e.g. in Belley AAC 2016 60 4342, 3 log reduction in 24h at $32 \mu\text{g mL}^{-1}$) and our data shows greater membrane

effects than dalbavancin. As such, we have modified our discussion and conclusions to suggest that the vancapticins have additional modes of action beyond peptidoglycan synthesis, without suggesting that they are similar to oritavancin. We have also noted that their time-kill properties still resemble those of vancomycin, not oritavancin.

Thus, this part remains weak in my view: link between depolarization permeabilization, and killing.

We believe that literature shows the link between membrane depolarization/permeabilization and killing speed for different classes of antibiotics is not particularly robust e.g. in a comparison of different antibiotics (Hobbs J Antimicrob Ther 2008 62 1003) the viability of *S aureus* after exposure to nisin, tetracycline, moxifloxacin and daptomycin showed dramatic differences between correlations of viability / membrane potential / cytoplasmic leakage over time.

However, we believe that our data demonstrate that the vancapticins do have membrane interactions causing depolarization and permeabilization, and that this evidence supports the hypothesis that these additional interactions lead to their enhanced potency compared to vancomycin.

2. All the mechanistic studies are presented in the discussion, which is very unusual. The description of the data should be in the results section, and the discussion should consider the data as a whole to give an interpretation. As it is now, data discussion is rather limited

We have restructured the manuscript to include the mode of action studies within the results section, as requested. The discussion section has been expanded to summarise the key findings but kept brief, in line with the journal guidelines "Discussion should be succinct and may not contain subheadings".

3. Caption to figure 6 is very obscure. Panel c is a model and I do not see how it illustrates the antagonism by free ligand. This bring me back to my comment to the original submission. This panel is a model for a possible interaction but NOT a ligand-antagonism ASSAY, which by definition, should correspond to experimental data. I apologize if I was unclear..

We thank the reviewer for clarifying their concerns. We have removed panel c from the Figure to avoid confusion.

4. Minor comment: the source of new products has both been added in the material section, if I am correct.

Sources of oritavancin and dalbavancin (and telavancin and daptomycin) now added.